# Lymphotropic Viruses EBV, KSHV and HTLV in Latin America: Epidemiology and Associated Malignancies. A Literature-Based Study by the RIAL-CYTED

**DOI:** 10.3390/cancers12082166

**Published:** 2020-08-04

**Authors:** Paola Chabay, Daniela Lens, Rocio Hassan, Socorro María Rodríguez Pinilla, Fabiola Valvert Gamboa, Iris Rivera, Fuad Huamán Garaicoa, Stella Maris Ranuncolo, Carlos Barrionuevo, Abigail Morales Sánchez, Vanesa Scholl, Elena De Matteo, Ma. Victoria Preciado, Ezequiel M. Fuentes-Pananá

**Affiliations:** 1Multidisciplinary Institute for Investigation in Pediatric Pathologies (IMIPP), CONICET-GCBA, Molecular Biology Laboratory, Pathology Division, Ricardo Gutiérrez Children’s Hospital, C1425EFD Buenos Aires, Argentina; paola_chabay@yahoo.com.ar (P.C.); elenadematteo@gmail.com (E.D.M.); preciado@conicet.gov.ar (M.V.P.); 2Flow Cytometry and Molecular Biology Laboratory, Departamento Básico de Medicina, Hospital de Clínicas/Facultad de Medicina, Universidad de la República, CP 11600 Montevideo, Uruguay; daniela.lens@gmail.com; 3Oncovirology Laboratory, Bone Marrow Transplantation Center, National Cancer Institute “José Alencar Gomes da Silva” (INCA), Ministry of Health, 20230-130 Rio de Janeiro, Brazil; chassan@inca.gov.br; 4Department of Pathology, University Hospital, Fundación Jiménez Díaz, 28040 Madrid, Spain; smrodriguez@quironsalud.es; 5Department of Medical Oncology, Cancer Institute and National League against Cancer, 01011 Guatemala City, Guatemala; cfvalvert@gmail.com; 6Department of Hematology, Salvadoran Institute of Social Security, Medical Surgical and Oncological Hospital (ISSS), 1101 San Salvador, El Salvador; irisarivera@icloud.com; 7Department of Pathology, National Cancer Institute—Society to Fight Cancer (ION-SOLCA), Santiago de Guayaquil Catholic University, Guayaquil 090615, Ecuador; fuadhuamangaraicoa@gmail.com; 8Cell Biology Department, Institute of Oncology “Angel H. Roffo” School of Medicine, University of Buenos Aires, C1417DTB Buenos Aires, Argentina; smranuncolo@gmail.com; 9Department of Pathology, National Institute of Neoplastic Diseases, National University of San Marcos, 15038 Lima, Peru; carlos.barrionuevo28@gmail.com; 10Research Unit in Virology and Cancer, Children’s Hospital of Mexico Federico Gómez, 06720 Mexico City, Mexico; abimor2002@yahoo.com.mx; 11Department of Integrated Genomic Medicine, Conciencia-Oncohematologic Institute of Patagonia, 8300 Neuquén, Argentina; vanesascholl@gmail.com

**Keywords:** Epstein–Barr virus, Kaposi sarcoma herpesvirus, human T-lymphotropic virus, lymphoma, Latin America

## Abstract

The Epstein–Barr virus (EBV), Kaposi sarcoma herpesvirus (KSHV) and human T-lymphotropic virus (HTLV-1) are lymphomagenic viruses with region-specific induced morbidity. The RIAL-CYTED aims to increase the knowledge of lymphoma in Latin America (LA), and, as such, we systematically analyzed the literature to better understand our risk for virus-induced lymphoma. We observed that high endemicity regions for certain lymphomas, e.g., Mexico and Peru, have a high incidence of EBV-positive lymphomas of T/NK cell origin. Peru also carries the highest frequency of EBV-positive classical Hodgkin lymphoma (HL) and EBV-positive diffuse large B cell lymphoma, not otherwise specified (NOS), than any other LA country. Adult T cell lymphoma is endemic to the North of Brazil and Chile. While only few cases of KSHV-positive lymphomas were found, in spite of the close correlation of Kaposi sarcoma and the prevalence of pathogenic types of KSHV. Both EBV-associated HL and Burkitt lymphoma mainly affect young children, unlike in developed countries, in which adolescents and young adults are the most affected, correlating with an early EBV seroconversion for LA population despite of lack of infectious mononucleosis symptoms. High endemicity of KSHV and HTLV infection was observed among Amerindian populations, with differences between Amazonian and Andean populations.

## 1. Introduction

Neoplasms of an infectious etiology account for about 16% of all cancers, which amounts to about two million cases per year, considering virus-, bacteria- and parasite-derived cancers. Interestingly, this number is significantly higher for developing countries, in which it can be as high as 30%, while in highly industrialized countries, such as the US, it can be as low as 5% [1]. The bases for this difference are not clear, but it may be due to the prevalence of the oncogenic infectious agents, or to additional co-factors causally linked to the infectious neoplasms.

Virus-derived cancers almost always originate from established chronic viral infections, because the mechanisms of viral persistence in the infected host are compatible with oncogenesis. Indeed, all human oncogenic viruses express proteins, and/or non-coding RNAs with the capacity to transform cells in culture and induce cancer in transgenic animals [2]. Viral oncogenes tend to enhance cell proliferation and survival, aiming to maintain the pool of infected cells during persistent infections. The International Agency for Research on Cancer (IARC) acknowledges seven viruses as direct human oncogenic agents: the Epstein–Barr virus (EBV), Kaposi sarcoma herpesvirus (KSHV), human T cell lymphotropic virus type 1 (HTLV-1), high risk human papillomaviruses (HPV), hepatitis B virus (HBV), hepatitis C virus (HCV) and Merkel cell polyomavirus (MCPyV) [3]. Although the human immunodeficiency virus (HIV) is also causally associated with several neoplasms, it is as an indirect oncogenic agent, due to the immunosuppression it imposes upon the infected host.

The Epstein–Barr virus (EBV) was the first oncogenic virus discovered 56 years ago by Dr. Anthony Epstein and Dr. Ivonne Barr. EBV was initially observed in samples of Burkitt lymphoma (BL) coming from the equatorial Africa, which represents the perfect example of the unequal geographical distribution of neoplasms of infectious origin. While EBV is responsible for close to 100% of the BL originating in this region, EBV only accounts for about 20% of all BLs in developed countries (see below). We later understood that whilst EBV is evenly distributed around the world, the African BL is also associated with repetitive infections with Plasmodium falciparum, an important co-factor of this neoplasm and a parasite endemic to this area. Today, BL is classified within three distinct clinical types, endemic (malaria- and EBV-associated), sporadic (derived from areas in which malaria is not holoendemic) and immunodeficiency-associated [4].

The world distribution of the oncogenic viruses varies significantly, while most adults are already infected with EBV, HPV and MCPyV indistinct of the geographic region, the others tend to be more prevalent in specific populations [2]. Even today, despite the global means of transportation and increased immigration that have allowed a more heterogeneous mix of populations, the prevalence of KSHV and HTLV-1 infection is still restricted to particular geographical areas, implying mechanisms of viral persistence in the population that are not explained by mere socioeconomic factors, but in which genetic susceptibilities, ethnic origin, culture and the prevalence of co-factors may be critical (see below). Like EBV, KSHV and HTLV-1 are associated with lymphoid neoplasms [2]. While EBV mainly infects and persists in B cells, KSHV and HTLV-1 persist in B cells and T cells, respectively, and, as such, they have been associated with B and T cell lymphomas [2].

Latin America (LA) comprises the land from Mexico to Argentina and the Spanish speaking Caribbean, countries with a complex mix of geographies, climates, politics, cultures, ethnicities and different levels of socioeconomic development, and, in which, a high prevalence of oncogenic viruses, acute tropical diseases and malnourishment collide. Early epidemiological studies documented a high seroprevalence of KSHV and HTLV-1 in some regions of LA, and to this day, it is common to find in the scientific literature that these viruses are endemic to LA (see below). However, the loco-regional estimation of their prevalence and induced morbidity remains poorly known. The RIAL-CYTED harbors a multidisciplinary Ibero-American network of clinical and basic researchers created to form a platform of multi-center cooperation focusing on increasing our knowledge of lymphoma, particularly for more underdeveloped or developing regions. This network aims to improve the diagnosis and prognosis of these neoplasms throughout LA by way of homogenizing its identification and classification. With that purpose, in this review, we sought to systematically organize and analyze the literature related to the lymphotropic and lymphomagenic viruses EBV, KSHV and HTLV-1, in order to better understand their loco-regional distribution and the risk our population carries in terms of developing lymphoma.

To this end, we explored three web search engines, PubMed, Google Scholar and SciELO, to access all related scientific publications in English, Portuguese and Spanish. Searches were done with the name of each LA country plus the name of each virus individually: Epstein–Barr virus or EBV; KSHV, KSV, HHV8 or Kaposi sarcoma virus; HTLV-1, ATLV or human T-cell lymphoma virus 1. All collected publications were read and classified according to the content; sero-epidemiological studies of KSHV and HTLV-1 viruses were considered, in order to discuss their association with lymphoma in specific regions, while EBV sero-epidemiological studies were discarded, since it is well known that there is a worldwide high prevalence in both developed and developing countries. In case reports and when series of lymphomas were included, we assessed the methods of viral diagnosis, prioritizing those studies in which the presence of the virus was addressed in the lymphoma sample by means of immune or molecular methods. The search was carried out from February 2019 to December 2019, and, although we aimed to include all papers found, regardless of the date of publication, many journals do not have online versions of the documents prior to the 1990s. Moreover, many old studies did not comply with more recent lymphoma classification. The results are disclosed in the next sections for each particular virus.

## 2. Epstein–Barr Virus

EBV is a human gamma-1 herpesvirus usually persisting as a harmless passenger; its growth-transforming ability is linked to a range of lymphoproliferative lesions and malignant lymphomas [5]. EBV-associated lymphomas vary according to the geographic location, age, sex, genetic background and socioeconomic condition [6]. Additionally, the age of primary infection varies substantially worldwide, correlating with socioeconomic factors [7]. In underdeveloped and developing populations, EBV infection is acquired at a young age and is usually asymptomatic. A delay in acquiring primary infection until adolescence or young adulthood, which usually occurs in more developed countries, can manifest as infectious mononucleosis (IM) in 25–75% of the late infected persons [8].

EBV infection has been associated with the following lymphomas in addition to BL: Hodgkin lymphoma (HL), lymphomas in immunosuppressed individuals (post-transplant lymphoproliferative disorders (PTLD) and HIV-associated lymphoproliferative disorders and T-cell and NK-cell lymphomas) [3]. Furthermore, the last WHO Classification of Tumors of Hematopoietic and Lymphoid Tissues included two new entities specifically associated with EBV: EBV-positive diffuse large B-cell lymphoma (DLBCL) not otherwise specified (NOS) and systemic EBV-positive T-cell lymphoma of childhood [9]. Since the incidence of HIV- and transplant-related lymphoma is more informative of the HIV prevalence or the number of transplanted patients than of the EBV distribution, we will not cover them in this review. See Table 1 for the studies considered and the frequency of EBV association.

### 2.1. EBV-Associated Lymphoid Neoplasms in Latin America

#### 2.1.1. Hodgkin Lymphoma

There are several series of lymphomas reported in this region, particularly at the end of last century, which, however, were only classified as Hodgkin and non-Hodgkin (NHL), and which obviously did not follow the latest WHO classification. HL has a bimodal age distribution, there is an early peak occurring in adolescents and young adults, and a late peak after 50 years of age in industrialized countries. Developing countries also exhibit a bimodal distribution of the disease, but the early peak starts before adolescence [121]. In the US, pediatric HL shows the highest incidence in adolescents between 15–19 years of age, while developing countries present similar incidences than the US for adolescents, but also exhibit a marked augmented incidence in young children [122,123]. In developing countries, classical HL (cHL) with an early onset (14 yrs or younger) shows high EBV association, more often of the mixed cellularity (HL-MC) subtype. A strong male to female predominance is also observed, particularly in the group younger than 5 yrs, in which the ratio is 5:1. HL in adolescents and young adults displays lower EBV association, nodular sclerosis subtype (HL-NS) predominance and affects male and females almost equally [122].

We found 36 studies of HL in LA, including pediatric and adult series from 12 countries. Pediatric series from Argentina exhibited a profile compatible with early cHL (i.e., younger ages, EBV positivity of ~50% and HL-MC as the predominant subtype) [20,25,26,27], while adult HL displayed a 31% of EBV-association, with similar frequencies of both NS and MC subtypes [22]. On the other hand, in the Southeast of Brazil, EBV positivity was 40–50%, HL-NS was the predominant subtype, and there was a smooth peak between adolescents and young adults [38,41,42,43,44,45,47,52,58,60,66,67,68,69,70,73], which may suggest a transition state in the epidemiology of the disease presentation between the ones observed in developing and developed areas. Indeed, one study in the most affluent Brazilian State (Sao Paulo) reviewed 817 cases of cHL over 54 years (1954–2008), describing that EBV-positive cases showed a decrease from 87% to 46% during the time of the study, with a remarkable decrease in young adults (85% to 32%) [32]. Nevertheless, HL-NS was still the predominant subtype in all periods. On the contrary, HL in the North of Brazil exhibits high EBV association (87%), HL-MC and young age predominance [61]. These results support the hypothesis that the socioeconomic level may determine the presentation features of this neoplasm, also highlighting the role of EBV as an HL driver that is also influenced by socioeconomic factors. Figure 1A–D shows examples of EBV positive HL-NS and HL-MC.

There is only one large pediatric cohort of 42 cases published from Mexico, with a median age of 5 yrs at diagnosis, and in which a male predominance (2.5:1) was observed [78]. EBV positivity was found in 76.1% of the cases and the HL-MC subtype was the most predominant (71.4%). Other studies from Mexico mostly include young and older adults, with scarce inclusion of children and adolescents, and in which EBV association frequencies vary between 61% and 77%. In those studies, a predominance of HL-MC and HL-NS subtypes was observed [82,87,88,89,118]. In these Mexican studies, a high EBV association was also observed for the lymphocyte depleted (HL-LD) subtype, with 91.2% positive cases [87,88,89].

In line with the above-described data, a 94% of EBV association was observed in pediatric HL in Peru [108], while in Colombia, EBV association varied from 60% in adults to 84% in children [93,94]. Another study from Honduras that included 11 children younger than 15 yo also found 100% EBV positivity [115]. Series of studies from the Caribbean and continental Central American countries: Puerto Rico, Ecuador, Costa Rica and the Dominican Republic found intermediate EBV association frequencies (50%, 55%, 36–40% and 64%, respectively) [90,91,110,116,117,118], which are closer to the frequencies observed in the more developed Argentina. These frequencies should be confirmed with more samples and up-to-date techniques.

#### 2.1.2. Burkitt Lymphoma

BL is a highly aggressive B cell lymphoma, characterized by the translocation of the MYC oncogene to the immunoglobulin loci. It affects mostly children and is also more predominant in males [124]. EBV-associated BL conforms almost 100% of endemic BL and around 20% of sporadic cases of North America and Europe. Other regions of the world, especially developing populations, exhibit intermediate frequencies, such as 50–70% in North Africa and Russia [3,125]. Pediatric BL in Argentina has an EBV association frequency closer to the sporadic subtype (25–47%), and the highest incidence of EBV-positive cases is in children younger than 5 yo [14,16,24,28]. As with HL, the frequency of EBV positive cases increase with latitude in Brazil, being ~60% in the South East [37,55,56,62,63,64], and 73–87% in North East [72,74]. A study including an extensive number of BL cases from five Brazilian geographic regions confirmed this trend, disclosing EBV association frequencies from 29% in the South to 76% in the North [53,54]. This pattern appears to represent a socio-geographic gradient, which might reflect social development, as well as other unknown environmental, ethnic or genetic factors [63]. A Brazilian study of 14 population-based cancer registries showed that the global age-adjusted incidence rate for pediatric BL does not differ significantly from the expected for a sporadic BL region. However, the incidence was elevated for BL children aged 1–4 years [126]. Given the association of EBV positive BL with young age, it is tempting to ascribe this elevated risk to the early EBV seroconversion, which is characteristic of the natural history of EBV infection in our geographic region [127]. Indeed, an EBV-associated BL inverse correlation with age has been shown in other studies from Brazil [56] and Argentina [14]. Unfortunately, we could not find reports on BL series from the Andean and Caribbean regions of LA, which, most probably, does not represent a low frequency of this lymphoma there. Although there are studies including BL from Mexico, they do not address EBV infection, and there is only one case report from a 63 yo male with an EBV positive intraoral BL [128].

#### 2.1.3. Diffuse Large B Cell Lymphoma

DLBCL is a highly aggressive neoplasm that can arise in almost any location of the body. It is very rare in pediatric patients and young adults, but is one of the most common NHLs in older individuals. Due to the median age of presentation of 70 yrs, it is usually associated with underlying immunosuppression. In the former 2008 WHO classification [129], EBV-positive DLBCL of the elderly (>50 years) was recognized as a provisional entity among the DLBCL subtypes. This provisional entity represented 5–11% of the DLBCL among immunocompetent East Asian patients [130], while in Western populations, the frequency was lower than 5% [81,131]. The reported series from LA have shown slightly higher EBV association frequencies (7% México, 9% Brazil, 13% Argentina and 14–28% Peru) [10,11,12,34,81,95,100,101] than the observed in Western countries [81]. LA patients are also younger than the ones described in other series [11,81]. Peru is particularly interesting, since it has the highest incidence of positive EBV DLBCLs, in addition to the incidence of DLBCL being the highest reported, accounting for up to 45% of all lymphomas [100,101,132]. In a Peruvian study of five cases, DLBCL of the gastrointestinal tract was consistently associated with EBV infection in elderly patients [98]. Remarkably, a series of cases of DLBCL of the palatine tonsils from Salvador de Bahia Brazil did not find an association with EBV [40]. Figure 1E,F shows an example of an EBV positive DLBCL.

In the 2016 WHO classification of lymphoid neoplasms, the age factor was eliminated from the association of EBV with DLBCL, leading to the substitution of the modifier “elderly” with “not otherwise specified” (EBV-positive DLBCL, NOS) [9]. Therefore, new studies free from the restraint of the age limit are needed, to better characterize the magnitude of this association, as well as the prognostic impact of EBV positivity in LA patients. DLBCL are also distinguished by their phenotype as germinal center or activated, with the former exhibiting an overall better survival rate. No differences were found in a series of Argentinian cases with respect to EBV positivity in these two subtypes of DLBCL [11,95].

#### 2.1.4. T and NK Lymphoproliferative Disorders

Although EBV tropism is mainly of B cells and in healthy individuals exclusively localizes in B cells [133], the revised 2016 WHO classification recognizes the chronic active EBV infection (CAEBV) of T/NK cell type, the aggressive NK-cell leukemia, the systemic T-cell lymphoma of childhood and the extranodal NK/T-cell lymphoma, nasal type (ENKTCL), as well as a new provisional entity, the primary EBV-positive nodal T/NK-cell lymphoma [9,134]. These neoplasms represent a broad spectrum of diseases that occur with higher incidence in Asian populations, and, in which, CAEBV, the aggressive NK leukemia and the systemic T-cell lymphoma are more prevalent in children and adolescents, while ENKTCL mainly affects adults [9,134]. How the virus infects T or NK cells is still a matter of debate, but exists some evidence of a preferential tropism for T cells for the EBV-2 subtype [135,136].

CAEBV corresponds to a group of reactive LPDs associated with a heightened EBV infection lasting longer than IM, and with the potential to progress to a systemic lymphoma. The clinical picture is diverse and includes the indolent, localized cutaneous form hydroa vacciniform-like LPD (HVL-LPD), renamed by the revised 2016 WHO classification [9]. Several series of patients with HVL-LPD have been described in LA indicating a high incidence of this disease, mainly in countries with a large Amerindian population component, such as Mexico [77,84], Bolivia [84], Peru [97,103,105] and Ecuador [109]. In Brazil, two HVL-LPD cases were reported in children from Amazon indigenous tribes, supporting the ethnic bias of this disease [31,39]. In these series of patients, the association with EBV was virtually 100%. These studies have also greatly contributed to illustrate the clinical and pathological features of this disease in LA as an LPD with 70% of cases being of T-cell origin and 30% of NK origin. This scenario is more similar to Asian CAEBV, since, in the US, CAEBV is very rare, and also presents as a B cell LPD [137]. Although cellular and viral monoclonality has been proven in the majority of T-cell cases, the disease is today considered an LPD, with a high risk to progress into a systemic lymphoma [77], to reflect the diverse clinical spectrum of the disease presentation, from self-limited HV to HV-like lymphoma, and also to allow for more adequate therapeutic approaches, since patients respond well to immunomodulatory agents as first line of treatment [77,103,105].

Extranodal NK/T cell lymphoma (ENKTCL) is an EBV-positive lymphoma more commonly derived from NK cells. Similar to CAEBV lymphoproliferations, this type of lymphoma has been described mainly in East Asia and LA, in which the ethnic composition includes a high proportion of Amerindians. EBV infection is also confirmed in virtually all cases. Series of cases have been reported from México [83], Peru [104,106], Chile [104,106,112,113] and Guatemala [120]. In Brazil, ENKTCL usually occurs as isolated cases [71]. Recently, a large series was reported that included 122 cases with mostly adults (only three children) from all five Brazilian regions [46]. In this Brazilian cohort, the clinico-pathological characteristics of the neoplasm were similar to the ones described in patients from East Asia and other American countries, in which the disease is considered endemic. No ethnic data was recorded for the patients included in the study. In a Brazilian unicentric study that included lymphomas involving the midline facial region, 16 were of T/NK cell origin and nine were of B cell origin (*n* = 25). Remarkably, no ethnic differences were found between the patients with T/NK or B cell presentation [71]. Figure 2 shows an example of an ENKTCL.

## 3. Kaposi Sarcoma Human Herpesvirus

KSHV is a gamma-2 herpesvirus closely related to EBV, also called human herpesvirus 8 or HHV8. KSHV is the etiological agent of Kaposi sarcoma (KS), multicentric Castleman disease (MCD) and primary effusion lymphoma (PEL). The revised WHO classification of hematopoietic and lymphoid tissues also included two entities associated with KSHV, the HHV8-positive diffuse large B-cell lymphoma, NOS and the HHV8-positive germinotropic lymphoproliferative disorder. KSHV-associated neoplasms usually develop in severely immunocompromised patients, and, as such, KS is, in many countries, an AIDS defining marker. The incidence of KS has sharply diminished after the introduction of anti-retroviral therapy, but has since stabilized in many countries [138]. In regions in which the virus is endemic, such as in Sub-Saharan Africa, HIV negative KS (classic KS or cKS) is one of the most common neoplasms in men [139].

MCD and PEL are extremely rare neoplasms. There are two main presentations of Castleman disease (CD), one in which reactive B cells only affect a single group of close lymph nodes, denoted as unicentric (UCD) and MCD, in which more than one group of lymph nodes are reactive. CD is also classified according to its microscopic morphology as hyaline-vascular (usually UCD), and plasma cell and plasmablastic (usually MCD). CD is not a true lymphoma, but rather a polyclonal B cell expansion and is thus considered an LPD. MCD is associated with HIV and KSHV co-infection, and if HIV is uncontrolled, MCD can have an aggressive clinical course and progress to an entity with cellular and viral monoclonality, in which it resembles a DLBCL [140]. On the contrary, PEL is an aggressive monoclonal B cell lymphoma with a poor clinical outcome, regardless of a down the line HIV control. PEL usually presents with lymphoid effusions in cavities, such as the pleura, peritoneum and pericardium. Post-transplant LPD and germinotropic LPD are also very rare and recently recognized [141,142], therefore, we could not find studies about them. Of interest is that in about 80% of PELs arising in AIDS patients, the tumor cells are co-infected with KSHV and EBV, a characteristic in common with germinotropic LPD [140].

### KSHV Prevalence and Morbidity in Latin America

To our knowledge, there is no routine screening for KSHV infection in blood banks in LA countries, nor in the rest of the world, and the World Health Organization and the World Federation of Hemophilia do not recommend it [143]. Thus, one problem when assessing the incidence of KSHV infection is that there is not a reference standard assay, and comparative studies have often demonstrated great disparity [144]. Therefore, comparing different studies may be inaccurate. Blood donor-based studies from Brazil illustrate this point (see Table 2), while some studies from Sao Paulo report a prevalence of 3.8–4.6% [145,146,147], another found 26% [148]; albeit, 1.1% positives were found among health workers and 20.4% in HIV/AIDS patients from the same city [149]. A screening of blood donors from the Amazonian region also found a high prevalence of 31.3% [150]. Pediatric studies found 7.6% positivity in Sao Paulo, 6.8% in Campinas and 12% in children of low-income families from Belem Pará [151,152,153]. Figure 3 summarizes studies about the KSHV prevalence among LA countries.

Argentina reports a prevalence of 3.6–7.4% [146,161,162,163], higher for San Salvador Jujuy in the North West of the country (12.3%), a region with a high population of immigrants and indigenous [161]. Higher prevalence for females than males was reported in two Brazilian studies, one pediatric (14.9:1) and one adult (3.7:1) [145,151], while in San Salvador Jujuy, the opposite was observed, with a male to female ratio of 3:1. More limited studies from Colombia (*n* = 25), Chile (*n* = 300) and Peru (*n* = 128) found 4%, 2.6% and 56.2% positivity, respectively [146,168,180]. In Peru, high endemicity was also challenged by a different study, in which the MSM high risk group was analyzed (*n* = 497), finding a seropositivity of 42.4% [181]. However, Peru is among the few countries from LA with multiple reports of cKS [182,183], including a large cohort of 126 cases that included pediatric cKS [184]. There are not large prevalence studies from Mexico, Central America and the Caribbean countries, a small screening of blood donors from Cuba (*n* = 171) found 1.2% KSHV positives [176]. Mexico reports KS as the most common neoplasm in AIDS patients, particularly before HAART [185,186]. For KSHV studies based on risk groups, see Table 3.

LA harbors distinct ethnic groups differentially distributed among different geographic regions in each country. Most urban populations are formed by mestizos and a minority of Caucasians, while some rural regions concentrate pure Amerindians or Afro-descendants. KSHV hyper endemicity is reported among LA indigenous populations and a hyper-endemicity area (30–100% prevalence) is reported among tribes inhabiting the South American lowlands of the Amazonas and the savannah (Figure 4 maps the prevalence of KSHV among Amerindian populations). For instance, Amazonian Amerindians from Ecuador were found 24% (the Siona, *n* = 41) to 100% (the Huaorani, *n* = 38) seropositive [195], while in Brazil different studies have found 56.8–79.1% among Amerindians from the Pará state (*n* = 339–1270) [150,196,197,198] and 50–83% (*n* = 110) among Amerindian tribes from transitional zones in central Brazil from Amazonian to the savannah. On the other hand, 0–14% positivity has been found in Amerindians from the South of Brazil (*n* = 85) [198]. Chacó Amerindians (*n* = 55) from Bolivia were 22% positive and 45% were positive in Paraguay [198].

Although our main aim was to document the incidence of lymphomas mediated by tumor viruses in LA, there are no such studies for KSHV in many LA countries, with most studies being about the prevalence of KS in HIV/AIDS patients. Additionally, because many of the reports are from the 1980s and 1990s, the studies only addressed the overall frequency of lymphomas, and lymphomas were only classified as Burkitt, Hodgkin and non-Hodgkin. Thus, both MCD and PEL seem more common after HAART, but they may have been under-diagnosed in early studies. Furthermore, KSHV-associated neoplasms most often are reported based on their histological and clinical characteristics, lacking confirmatory tests for viral infection. KSHV-associated lymphomas are reported from Brazil, Argentina and Peru. We found five cases of MCD in Brazil, mostly of the plasmablastic variety, and one mixed plasmablastic and hyaline [154,155,156,157,158]. Although KSHV infection was not assessed in all cases, four cases were in HIV infected individuals and in three cases MCD co-presented with KS. There was also a case of PEL in an HIV positive patient [159], and a multiple myeloma patient with a bone marrow KSHV PCR positive [160]. We found three KSHV positive MCD from Argentina, two PEL and two KSHV positive DLBCL, all in HIV positive patients [13,165,166,167]. One of the DLBCL was also positive to EBV infection [164]. On the other hand, a screening of a series of cases of B cell lymphomas arising in HIV/AIDS patients (*n* = 49) did not find any KSHV positive supporting the rarity of these kind of neoplasms [199]. Peru reported two cases of UCD hyaline that were KSHV positive by IHC [169]. We found only one PEL and three MCD reported from Mexico [170,172,173], and two PEL and one MCD from Cuba [179,187]. Figure 5 shows KSHV positive PEL and MCD.

There are five major KSHV genotypes described (A, B, C, D and E), mainly based on the hypervariable regions of the oncogenic K1 gene. Interestingly, in the series of studies from LA, while genotypes A, B and C [167,177,190,200,201,202] were described in neoplasms or HIV positive cases, subtype E has been the sole genotype found in Amerindians from Ecuador, Brazil and French Guiana [190,195,197,203,204,205]. Genotypes A and C are prevalent in Europe and North America, while genotype B is more prevalent in individuals with African ancestry [195]. Thus, one plausible explanation for the low incidence of KSHV-induced neoplasms (including cKS) among South American Amerindians is a low oncogenicity of the KSHV genotype E. Indeed, only two KS reported in Peru shared genotype E [204], while KSHV A and B genotypes have been isolated from Brazilian and Argentinian patients with European ancestry [201,206]. To our knowledge, the tumorigenic activity of the genotype E K1 has not been tested in vitro or in animal models, and it is not clear why KSHV would become more attenuated in South American Amerindians, but not in other populations with an equally prevalent KSHV infection, such as those from Uganda or Zambia.

## 4. Human T-Cell Lymphotropic Virus

HTLV belongs to the family Retroviridae, genus Deltaretrovirus, and, as such, its genome consists of two identical copies of positive sense single stranded RNA that during its life cycle is converted into double stranded DNA and inserted into the DNA of the human host cell. This integrated form is referred to as the provirus, and HTLV predominantly exists in the host as a cell-associated provirus. Indeed, infected cells hardly produce any virus and plasma viral load is usually undetectable. There are two main types of HTLV infecting humans, HTLV-1 that preferentially infects CD4+, but also CD8+ T-lymphocytes, and HTLV-2 that preferentially infects CD8+ T-lymphocytes. Two other HTLV types have been described in Africa, but today, it is not clear whether HTLV-3 and HTLV-4 can be transmitted from human to human or are exclusively zoonotic, and no clinical disease has been associated with these two novel isolates thus far [207].

HTLV-1 and -2 are very similar in terms of genome structure and replication patterns, but they are strikingly different in terms of clinical impact and geographical distribution. HTLV-1 is present throughout the world with regions of high endemicity together with low prevalence in neighboring zones. Major endemic regions include Southwest of Japan, LA and some areas of West Africa [208,209]. Ten to twenty million people are estimated to be infected with HTLV-1. However, the world prevalence of HTLV-1 is largely unknown, since most published studies are performed on blood donors and there are not population-based studies. The epidemiology and pathogenic potential of HTLV-2 infection are less well characterized. HTLV-2 is present in many world areas, and high HTLV-2 seroprevalence has been reported in indigenous populations in Africa and Amerindians from Central and South America, as well as among IVDU in Europe and North America [209]. Moreover, six genotypes have been found for HTLV-1 based on the genomic sequences of the long terminal repeats, A (also called the Cosmopolitan strain, because of its worldwide distribution) to G (the rest of them with a somehow more restricted geographical distribution) and five for HTLV-2, A to E [210].

HTLV-1 infection has been associated with adult T cell leukemia/lymphoma (ATLL) and a subacute myelopathy known as tropical spastic paraparesis or HTLV-1-associated myelopathy (TSP/HAM). Although most HTLV-1 carriers are asymptomatic, about 5% of them will develop clinical complications, including TSP/HAM and ATLL. Both diseases are severe and produce progressive incapacitation or death. In addition to ATLL and TSP/HAM, infection by HTLV-1 also leads to a broad spectrum of inflammatory diseases with significant morbidity, such as uveitis, arthritis, myositis, infective dermatitis, Strongyloides stercoralis infection and crusted scabies [209,211]. The mechanisms leading to them are currently unknown. On the contrary, and in spite of sharing 60% of genetic homology, no disease has been consistently associated with HTLV-2 infection, although sporadic cases of subacute myelopathy or hematological malignancies have been reported [209].

### HTLV Prevalence and Morbidity in Latin America

When assessing the viral prevalence, we found that the majority of studies are based on immunological methods that do not discriminate between HTLV-1 and HTLV-2 (referred to as HTLV). The Caribbean basin is usually considered endemic to HTLV infection, perhaps because early 1980s studies documented several cases of ATLL from black immigrants from the Caribbean islands [212,213]. There are several studies from Cuba and Dominican Republic that tried to confirm that high endemicity and morbidity. Surprisingly, three sero-epidemiological studies carried on Cuban blood donors found 0.25%, 0.27% and 0.99% HTLV positivity, even though they included a significant number of black Cubans [214,215,216]. Dominican Republic and Puerto Rico report 2.5% and 0.2% positivity in blood donors, respectively [217,218,219]. Higher prevalence has been reported for risk groups. A retrospective study from 1996 to 2007 from two sentinel hospitals from Havana and Matanzas Cuba that registered all lymphoid neoplasms, reported only five ATLL cases [220]. On the contrary, in spite of multiple reports about TSP/HAM in Dominican Republic and Puerto Rico [218,221,222,223], we could not find a single ATLL case. Although it is not clear which is the most prevalent subtype, HTLV-1 is almost exclusively the one reported [216,220,224,225]. Thus, the low incidence of ATLL may be due to the low prevalence of HTLV infection, or the scarcity of other needed co-factors. For a summary of data see Table 4.

Zero to 0.39% HTLV positivity and no ATLL were found from Mexico [226,228,342,343]. Interestingly, in those studies in which subtypes were identified, HTLV-2 was the only or the most prevalent one [344,345,346,347]. In agreement, HTLV-2 was highly prevalent in IVDUs (22%) [346]. From Central American countries, 5% HTLV-1 positivity has been found in the metropolitan areas of Panama City and Colon [229], and 0.24% and 0.68% HTLV-1 positive cases in Nicaragua and Costa Rica, respectively [235,348]. Honduras exhibited significant differences between regions, namely a 0.3–1.1% prevalence reported in the Central capital of Tegucigalpa and in the North West of the country, compared to a higher prevalence observed in coastal cities, particularly among black natives (17.7%) and non-mestizos (8.1%) [232,233,349]. High prevalence of 8.3–9% has also been reported among indigenous populations of Panama, all carrying HTLV-2. Although multiple cases of TSP/HAM have been reported, mostly associated with HTLV-1 infection [349,350,351,352,353], only five ATLL cases have been reported in Central American countries, one associated with HTLV-1 infection and the others of an unknown etiology [230,231,234].

Screening for HTLV infection in blood banks is mandatory in several South American countries; Brazil has done it since 1993, Peru since 1998, Uruguay since 2000, Argentina since 2005, Chile since 2009 and Colombia since 2014. This has allowed for studies based on a large number of samples, and for a more factual estimation of HTLV prevalence. In Caracas, Venezuela a 0.2% HTLV positivity and a rate HTLV-1: HTLV-2 of 11:1 was observed [236]. Other similar studies also found low HTLV positivity in Maracay (0.58%) and in Caracas (1%), but higher in the Amazon region (13%) [237,238]. A study that included all Colombian departments found an overall seropositivity of 0.3%, with most departments below 1% and Chocó being the only exception with 6.28% [239]. Similar studies from several blood banks from Medellin observed 0.06–0.176% positivity, mainly for HTLV-1 [243,244]. We also found two studies conducted on blood donors from Peru, Arequipa (0.9% HTLV-1 positive) and Abancay (3.4%, unknown subtype), and one in pregnant women from Lima (1.7% HTLV-1 positive) [253,254,256]. Concerning Chile, 0.12–0.24% HTLV positivity was described, showing higher prevalence in the central region of Valdivia than in Concepcion in the South; together with an almost 3:1 female to male ratio and a 7:1 HTLV-1 to HTLV-2 ratio [265,274]. A 0.13% HTLV positivity was found in Uruguay, ranging from 0.2% in the North to 0.09% in the South [277]. Those studies that were not based on archival data taken from blood donors reported an HTLV seropositivity of 0.42% in continental Ecuador, 1.57% in the Galapagos Island [252] and 2.4–3.5% for Afro-descendants from the Esmeralda province [250,251], 4.3% HTLV-1 in Bolivia [354] and 0% HTLV positives [276] to 2.5% HTLV-2 in Paraguay [355]. Figure 6 summarizes studies about the HTLV prevalence among LA countries.

Multiple studies have addressed HTLV prevalence in Brazil and Argentina revealing regions of high endemicity. In the former, the highest prevalence was demonstrated in the North-North East region, in particular in Salvador de Bahia, ranging from 0.046% to 1.8% [303,306,308,309,310,312,313,319,320,356,357,358,359,360,361,362,363,364,365,366], with slightly higher prevalence in specific villages with African characteristics, such as Piaui (2.24%) [359], Taquarendi (3.85%), Junco (1.23%), Alegre (1.56%) [363] and Marajó Island (1–2.06%) [366]. In the South and South Eastern regions, HTLV prevalence ranged from 0.02% to 1.5% [302,303,305,318,321,322,358,367,368,369,370]. In urban Northern and Southern regions HTLV-1 prevailed over HTLV-2 [309,310,313,314,317,358]. The central region exhibited HTLV prevalence ranging from 0.02% to 1.6% [305,307,315,316,317,371,372,373,374], of which, for the Amazonian region, prevalence increased when indigenous villages were analyzed: 0.4–1.18% HTLV positivity in the city [375,376,377,378], whereas, in certain villages, the prevalence increased up to 13.88% [376,379,380,381,382,383]. In Argentina, most seroepidemiological studies were performed in the North and central regions, and an HTLV-1 endemic area was discovered in Jujuy, a North Western province. HTLV prevalence in central provinces ranged from 0% to 0.2% (0.028–0.26% HTLV-1; 0.007–0.03% HTLV-2) [281,282,284,285,287,288,289,290,291,292,294,295,296,384]. In contrast, this incidence is increased up to 9.8% of HTLV-1 in Jujuy [280,292,385,386], and to 31.5% when Jujuy family clusters were studied [387]. Surprisingly, other Northern Argentinean cities reported low HTLV positivity, namely 0.16–0.7% in Salta that borders with Jujuy [284,292], 0.6% in Formosa [292] and 0.007% in Misiones [295]. In the above-mentioned epidemiological studies, HTLV positivity seems enriched in populations with a high proportion of Amerindians, particularly for HTLV-2. Additionally, a higher prevalence of 17–19% was found among first generation Japanese immigrants, particularly those coming from the HTLV-1 endemic region of Kyushu [354,388]. Second and third generation Japanese-Bolivians were 6% HTLV-1 positive, closer to the native Bolivian population. HTLV prevalence has also been addressed in Japanese immigrants in Brazil (1.8–6.8%) and their descendants (0.73%) [361,389,390]. See Table 5 for HTLV prevalence in other risk groups.

While the most pathogenic HTLV-1 strain is more prevalent among urban mestizo, Afro-descendants and Caucasian populations, HTLV-2 seems to be more prevalent among some Amerindian tribes. Interestingly, different endemic zones for HTLV-1 and HTLV-2 infections have been described among different tribes of Colombian Amerindians [426,427,428,429], with a general higher HTLV-1 prevalence in tribes from the Andes and other highlands than in lowlands, and vice versa for HTLV-2. Likewise, the Peruvian Andeans such as the Quechua exhibit 2.8–5.1% HTLV-1 positivity and 0% HTLV-2 [430,431]. The exceptions of Amazonian communities with an almost exclusive HTLV-1 infection are the Shipibo-Konibo (1.9%), the Wayku (2.5%) and San Francisco (1.4%) [432,433]. In Chile, higher prevalence was also found in Amerindian tribes than in mestizo urban populations, with those from the highland Atacama region exhibiting the highest positivity with 6.5%, while Mapuches were 0.7–1%, Huilliches were 1.9% and natives from the island of Chiloe and Pitrufquen town in the South of Chile were 1.5% [434,435,436]. Additionally, in those studies in which the viral strain has been addressed, an almost exclusive presence of HTLV-1 was found [434,436]. HTLV-1 infection among Bolivian Amerindians was 5.3% in the Aimara and 4.5% in the Quechua [437]. Other Amerindian populations outside the high endemicity region of the Amazonia, such as Guaranies, Kayapo and Kraho tribes, showed HTLV prevalence of 5.7%, 33.3% and 12.2%, respectively [438,439]. It is worth mentioning that, while the HTLV-1 prevalence was below 11% in endemic regions, HTLV-2 prevalence was significantly higher [428,429]. An international study that analyzed Indians from Colombia, Venezuela, Bolivia, Brazil, Paraguay and Chile found HTLV-2 positivity of up to 57.9% in the Kayapo, Brazil, 34.8% in Alacalf, Chile and 16.4% in the Chaco, Paraguay [437]. Higher HTLV-2 positivity has also been demonstrated for Amerindians populations from Argentina, such as the Toba (10–36.4%), Gran Chaco (22%), Mataco (3%) and Mapuches (2%) [284,290,385,386,387,412,437,440,441,442,443]. Other Amerindians that exhibited an almost exclusive HTLV-2 prevalence were those from the Gran Chaco region of Paraguay (4–44%) [355,442], Boca Colorado (4.54%) and Galilea (2.38%) in Peru [443] and the Yaruro/Guahibo (24.8–61%) and the Pume (5%) in Venezuela [444,445,446]. Of note, ethnic groups with 0% positivity to both types have been described in Venezuela, Colombia, Ecuador and Bolivia [251,426,437]. Figure 4 maps the prevalence of HTLV-1 and HTLV-2 among the Amerindian populations.

There are multiple reports of ATLL in South America, mostly describing a disease with similar features to Japan ATLL, but in younger individuals. In Colombia, 19 HTLV positive ATLL cases have been reported, 12 confirmed as HTLV-1, while, in the others, the HTLV subtype was not investigated [246,247,248,249]. On the other hand, about 200 of TSP/HAM cases have been documented, with half of them confirmed to be in HTLV infected individuals [249,447,448,449,450,451]. It is also notable that many cases come from the Tumaco district, in which TSP/HAM was originally described and in which a 5.3% HTLV-1 positivity has been reported among adults [241]. Furthermore, 12 ATLL cases from Peru and an additional case of a Peruvian immigrant in Uruguay were described, all of them being fatal [257,258,259,261,262,263,279]. These ATLLs displayed common features, since half of them had gastrointestinal tract compromise, and half of them occurred in individuals younger than 50 yr, including a 20 yr female. There were also seven DLBCL developing in HTLV-1 seropositive patients, but in which, the presence of the provirus was not assessed [260]. The tumor cell also harbored EBV in three of these DLBCLs. As an HTLV endemic country, Peru reports large cohorts of TSP/HAM reflecting a common disease [452,453,454,455], and also cases in which HTLV infection correlates with infective dermatitis [456], including children in which neurological disorders co-exists with infective dermatitis [457] and Strongyloides stercoralis [458,459]. Fifty-four cases of ATLL have been reported from Chile [266,267,268,269,270,271,272,275]. Highlights are the early presentation, with medians of 50 yrs and 51 yrs for the largest cohorts [266,268], and an onset concomitant with or proceeded by TSP/HAM. It is also interesting to point out that all reported cases in large cohorts are of Caucasian origin (49/49) [266,268,270]. Barrientos A, et al. found HTLV positivity by PCR in 16/88 (18.2%) cases of hematological malignancies (15 HTLV-1 and 1 HTLV-2), including two of myeloid origin. The median age of diagnosis was 33 yrs (2–92 yrs), with 50% of HTLV positive patients being 30 yrs or younger. No ATLL assignation was given in this study [273]. Multiple cases of TSP/HAM are also reported from Chile [460,461,462,463], but, contrary to ATLL, TSP/HAM occurs in Caucasians, in mestizo and Chilean Amerindians. There were no reports of other HTLV-associated diseases. In Argentina, ATLL cases have been described in the highly endemic Jujuy, with five cases with confirmed diagnoses [298]. There are seven other ATLL cases reported (two originating in brothers) [300,301]. In addition, TSP/HAM cases have been described in Aymara Amerindians, also in Jujuy [464], and also in Caucasian patients from other non-endemic regions [465,466,467,468]. Figure 7 shows an example of an ATLL with a CD4+CD25+FOXP3+ regulatory T cell phenotype.

The number of ATLL and TSP/HAM cases reported from Brazil is in agreement with a country of high HTLV endemicity. For instance, ATLL cases have been reported in the urban areas of North East Brazil, by and large in the Bahía state that presents the largest prevalence of HTLV-1 infection [326,327,333,335,336,469,470,471]. Characteristics of these ATLL are that they displayed cutaneous involvement [323,324], or were associated with infective dermatitis (IDH) [336], hyalohyphomycosis [333], Strongyloides stercoralis [335] or Hodgkin-like features [332]. Remarkably, ATLL has also been described in 31 pediatric patients [472]. In the South of Brazil, most reports are from Rio de Janeiro, where ATLL features often included a primary cutaneous type presentation [330,340,394,406], 9% (of 195 patients) co-presented with TSP/HAM [340], and eight cases were in pediatric patients [328]. There are at least 27 articles describing TSP/HAM cases that also support a common disease. TSP/HAM incidence was increased in HTLV-1 and HIV positive individuals (18–30%) versus those who were only HTLV positive (1.93%) [473,474], and pediatric cases of TSP/HAM have also been reported [472]. HTLV infection was also associated with IDH [475] and other dermatologic [476,477,478,479,480], oral [481], ocular [482] and neurologic [483] complications. Although there are multiple reports of TSP/HAM from Venezuela, Ecuador, Uruguay and Paraguay, we could not find a single report of ATLL in these countries.

HTLV-1A has been found throughout South America [282,304,484,485,486,487]. On the other hand, Amerindians with high HTLV-2 prevalence almost exclusively carry the HTLV-2B genotype [442,445,446,488,489]. HTLV-2B was also the prevalent genotype in Mexico [345,420] and Panama [490], while HTLV-2A [491] and -2C [492] have been found in Brazilian Amerindians co-infected with HIV, and HTLV-2A in healthy Kayapo Amerindians [493]. This is important because, although published studies support a relative high incidence of ATLL in South American countries, the disease seems most prevalent among mestizo and Caucasian populations than among Amerindians, despite their higher HTLV-2 positivity.

## 5. Discussion

The association between EBV infection and lymphoma varies worldwide. The most compelling observation is the higher incidence of EBV-positive cHL in LA than in industrialized countries, and also the relationship between the age at which EBV primo-infection occurs and the onset of cHL. In underdeveloped and developing LA populations, earlier exposure to EBV in the face of a relatively underdeveloped immune response might be a predisposing factor for EBV-positive cHL. Socioeconomic improvement in Brazil turned into a decreased incidence from 87% to 46% in a 54 year period [32]. A similar picture emerges for EBV-positive BL, since this disease shows features of endemic BL in particularly underdeveloped regions, for instance, 76–100% in North Brazil [36,56], while it resembles more the sporadic form in the South of Brazil, Chile and Argentina [114]. Fewer driver mutations, especially among genes with roles in apoptosis, were demonstrated in EBV-positive BL from USA and Uganda, suggesting a specific BL phenotype, irrespective of geographic origin [494]. Particular pathogenic mechanisms may predispose against EBV-positive cHL and EBV-positive BL, such as ethnicity, endemic infectious agents, low age of seroconversion or differences in socioeconomic development.

EBV-associated T and NK LPDs represent a broad spectrum of diseases that occurs with higher incidence in Asian and LA populations. The incidence of these lymphomas in LA should grant EBV1 and EBV2 prevalence studies, particularly for Mexico and Peru, in which this lymphoma is highly prevalent. NKT LPDs often emerge within CAEBV disease in which infection of T- or NK- cells apparently occurred during primary infection. Genetic alterations are rare in CAEBV, but mutations in *DDX3X*, *KMT2D*, *BCOR*, *BCORL1*, *TET2* and *KM6A* have been identified [495]. Likewise, LA has shown slightly higher EBV association frequencies in DLBCL, particularly in Peru. Peru has also reported cases of systemic EBV-positive T-cell lymphoma of childhood [99]. Gene expression profiling shows that EBV-positive DLBCL NOS, is molecularly distinct from EBV-negative DLBCL in Western developed populations, with NFκB p50, STAT-3 and CD30 more commonly expressed in EBV-positive DLBCLs [496]. No genetic study has been conducted in DLBCL of Hispanic patients so far.

ENKTL represents around 23% of all T-cell lymphomas in LA (up to 40% in Mexico and 66% in Guatemala), compared with 4% to 5% in Europe and the United States. The clinical presentation is similar in all geographic regions; however, patients in LA tend to present with more advanced clinical stages [497]. In the ENKTCL arising in East Asia, genetic analyses have shown that activating mutations of the JAK-STAT pathway, such as in *JAK3* (5–35%), *STAT3* (6–27%) and *STAT5B* (2–6%) genes are characteristic of this lymphoma [498,499]. Other mutations include the RNA helicase DDX3X, the tumor suppressor gene *TP53*, as well as genes encoding proteins involved in epigenetic pathways (*MLL2*, *ASXL3*, *ARID1A* and *EP300*) [134]. In a study of 71 ENKTCL cases from three different centers in LA (42 from Mexico, 17 from Peru and 12 from Argentina), mutations affecting the JAK-STAT signaling pathway were identified in 27% of cases, being *STAT3* the most frequently mutated gene (22%). Mutations in *BCOR*, *DDX3X*  and *TP53* were also identified, but with different frequencies than in Asian cohorts [500]. These data indicate that ENKTCL shows a similar mutational landscape in LA and Asia.

With respect to KSHV, available reports based on healthy populations and risk groups make it possible to infer an LA prevalence higher than in the US and non-Mediterranean Europe but lower than endemic regions of Africa and the Mediterranean. With the exception of Chile, LA exhibits global frequencies above 3%, and particularly above 10% for Cuba, Colombia, Argentina (San Salvador Jujuy), Ecuadorian Afro-descendants and Peru. Indeed, large KS series (e.g., ≥ 50 cases) from Cuba, Colombia and Peru have been published [184,501,502], including a series of cases of cKS published before the HIV/AIDS era, and of pediatric KS [503]. KS has also been documented as the third most common neoplasm in Peru [504].

In Amerindian tribes in which the KSHV genotype E is endemic, viral transmission starts very early, with 35 to 65% seropositives by age 10 [150,196,197,505], supporting casual contacts between family members as an important via of viral dissemination. Indeed, close contacts of KS patients have higher KSHV prevalence than the general population (Table 3). A similar scenario is found for HTLV, with seroprevalences increasing with age [442,506]. A study of 104 mothers with TSP/HAM showed that 19–31% of their children were already HTLV seropositive [454], supporting an early mechanism of vertical transmission, such as breast feeding, which could explain the pediatric presentation of ATLL and TSP/HAM in LA HTLV endemic zones. On the contrary, among non-endemic populations, the high prevalence among MSM and FSW supports sexual transmission as an important route for KSHV dissemination (Table 5). However, the latter is not as clear for HTLV infection, since those studies based on people attending STD clinics (0.6–2.8%), FSW (below 5% in most studies) and MSM (2% and below) reported lower frequencies than those observed for KSHV. An important exception is HTLV transmission among IVDUs that is consistently high (7.8–20.7%) in LA countries, with a notable higher prevalence of HTLV-2, and also in blood transfused individuals (0.4–10%), supporting contaminated blood as an important source for viral transmission. It is worth mentioning a study from Venezuela, in which three of five organ recipients from an HTLV-1 positive multi-donor became seropositive [507].

While ATLL among Amerindians is very rare, TSP/HAM is commonly reported. TSP/HAM was first described in the Southern Pacific of Colombia in 1981 as a chronic non-hereditary spastic paraparesis, initially termed Pacific spastic paraparesis [508], later also observed in the Caribbean islands of Martinique and Jamaica and in HTLV endemic regions of Japan. TSP/HAM seems to be highly prevalent in LA, particularly for Afro-descendants, but also for mestizo and Caucasian populations [223,249,351,448,449,452,453,454,455,461,464,472,509]. Likewise, infective dermatitis, strongyloidiasis and many other dermatological, oral, ocular and neurological complications are usually found co-presenting with ATLL or TSP/HAM, or independent of those diseases in HTLV endemic regions of Peru, Chile and Brazil. Infective dermatitis and strongyloidiasis sometimes precede TSP/HAM or ATLL [248,325,456]. It is also worth mentioning that extensive series of pediatric TSP/HAM have been reported from Salvador de Bahia and Rio de Janeiro in Brazil and in Peru [457,472]. All these studies pointed out that South America represents a major endemic area for HTLV infection and associated diseases. Remarkably, Mexico did not report HTLV positive ATLL or TSP/HAM.

Reports of ATLL series were found only for Brazil and Chile (see Table 4), which makes it difficult to identify the unique features of the disease presentation in our population. Furthermore, most case reports lack molecular and serological methods to confirm HTLV infection. Nevertheless, during the review process those cases in which the neoplasm co-presented with TSP/HAM, infective dermatitis or strongyloidiasis, or with the characteristic flower cell, were assumed to be bona fide ATLL. However, absence of confirmatory tests may miss early ATLL, for instance, in the smoldering stage, in which there is no significant lymphocytosis. Still, there do not seem to be major clinical differences between the ATLL cases in LA and those from Japan, or from other parts of the world [340]. Perhaps the most distinctive feature is an early presentation. Iwanaga et al. reported that in Brazil the median age of presentation is 44 yrs, while in Japan it is 68 yrs [510], and, in spite of being named as an adult disease, there are pediatric cases of ATLL in Brazil [323,325,328,340,471]. The Chilean ATLL cohort also had a younger disease presentation than Japan, with a median age of 50 yrs [266]. Most studies also support a poor prognosis for ATLL, the study of 195 cases from Brazil found an overall survival of 9.2 months [340].

## 6. Conclusions

We observed a high incidence of EBV-positive lymphomas of T/NK cell origin for Mexico and Peru and of BL for North Brazil. Peru also carries the highest frequency of EBV-positive HL and DLBCL than any other LA country. The presentation of HL and BL was also earlier than for industrialized countries, coinciding with an early EBV primo-infection, in spite of there being no evidence of infectious mononucleosis in the region. High prevalence of KSHV and HTLV was found among Amerindian populations, with different distributions between low-land (e.g., the Amazons) and high-land (e.g., the Andes) tribes. In spite of this, there were only scarce reports of MCD, PEL and ATLL. However, Peru reports KS as its third most common neoplasm.

## Figures and Tables

**Figure 1 cancers-12-02166-f001:**
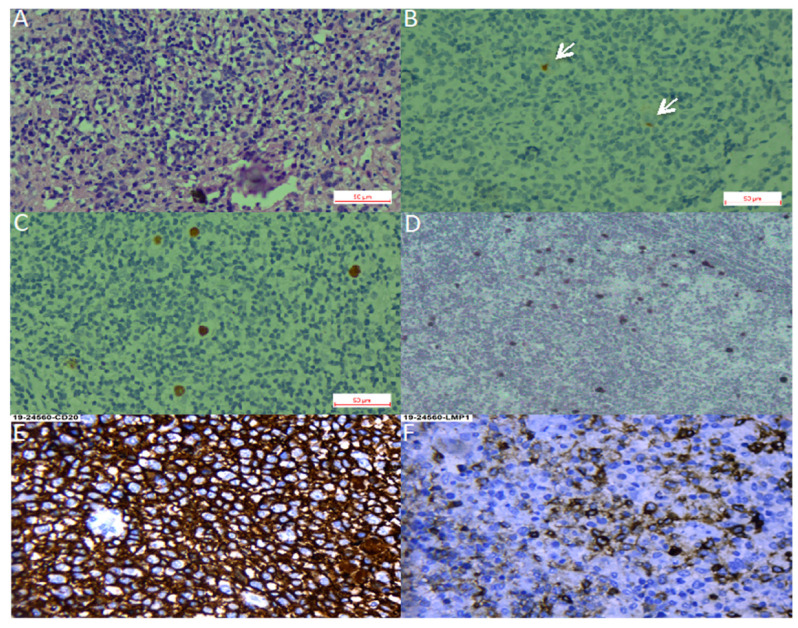
Hodgkin lymphoma. (**A**–**C**). Nodular sclerosis Hodgkin lymphoma. (**A**). Hematoxylin-Eosin staining. (**B**). CD30 staining with arrows pointing at positive Reed-Sternberg cells. (**C**). Epstein–Barr virus-encoded small RNAs (EBER) in situ hybridization. (**D**). EBER in situ hybridization of a mixed cellularity Hodgkin lymphoma, magnification 40×. (**E**,**F**). Diffuse large B cell lymphoma, magnification 40×. (**E**). CD30 staining. (**F**). LMP1 (latent membrane protein 1) staining.

**Figure 2 cancers-12-02166-f002:**
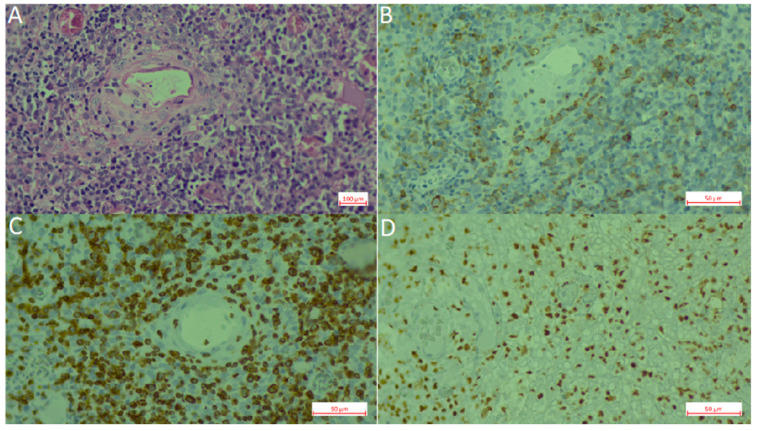
Extranodal natural killer (NK)/T-cell lymphoma, nasal type (ENKL). (**A**). Hematoxylin-Eosin staining. (**B**). CD56 staining. (**C**). CD3 staining. (**D**). EBER in situ hybridization.

**Figure 3 cancers-12-02166-f003:**
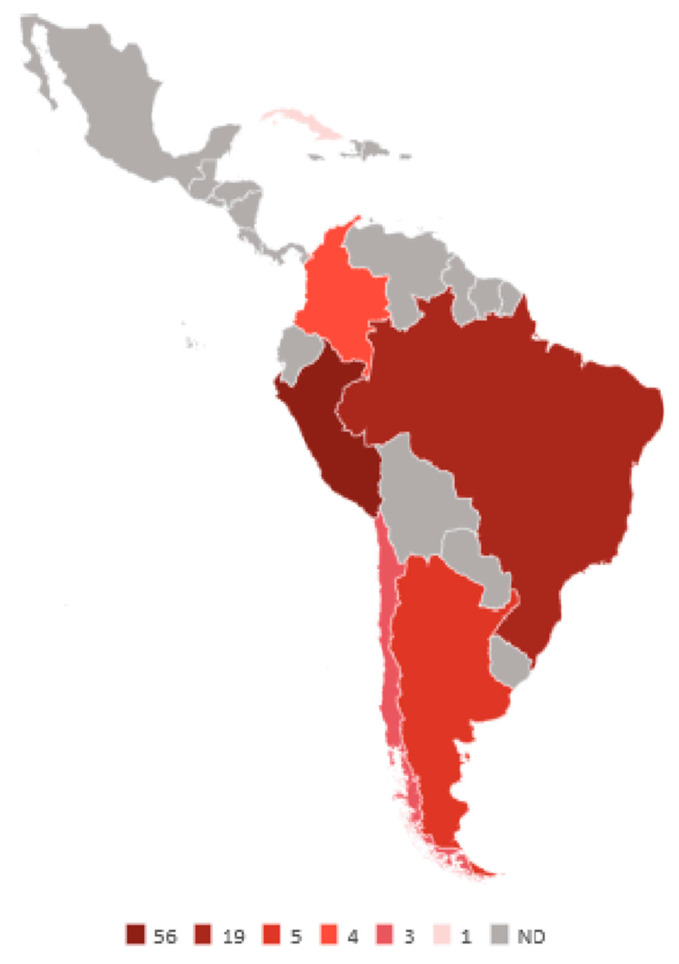
Kaposi sarcoma herpesvirus (KSHV) prevalence. Numeric values are in percentage. ND: not determined.

**Figure 4 cancers-12-02166-f004:**
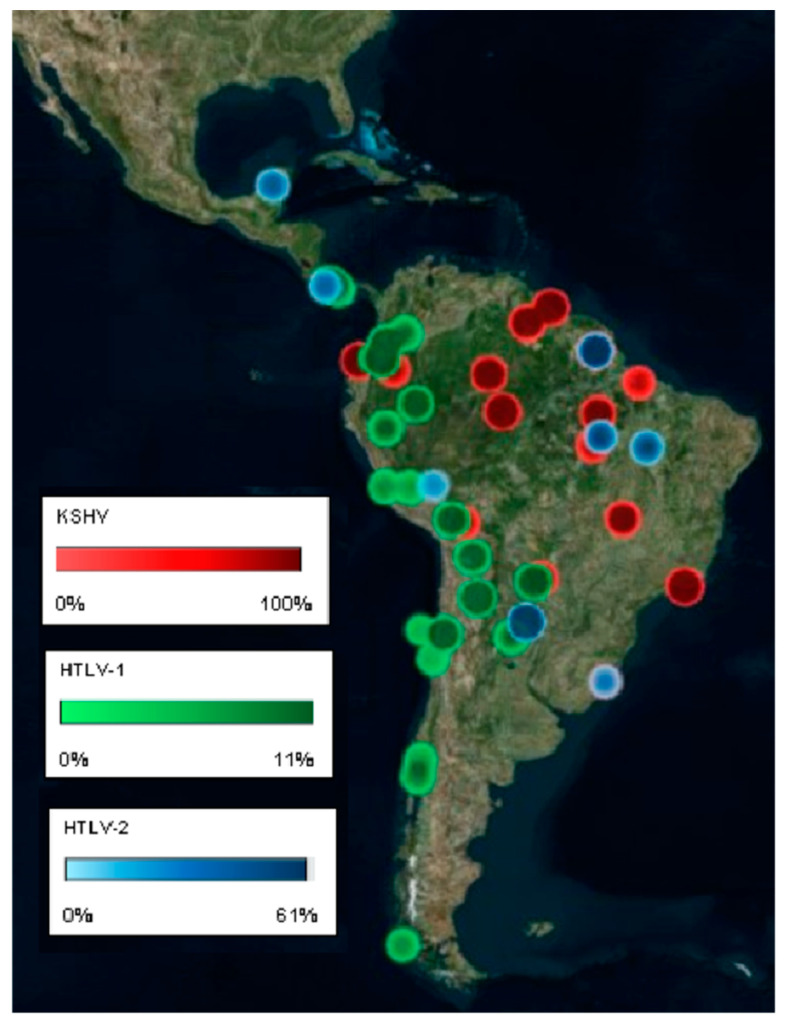
KSHV and human T-lymphotropic virus (HTLV) locoregional prevalence among Amerindian tribes.

**Figure 5 cancers-12-02166-f005:**
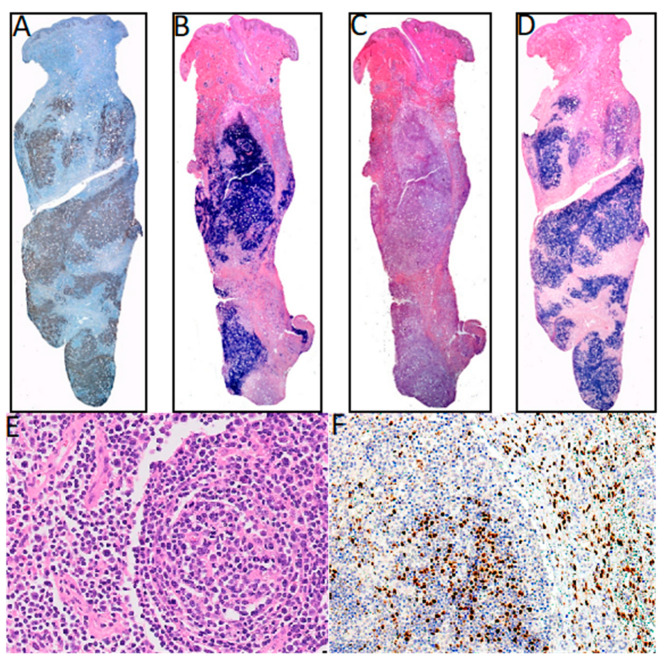
KSHV associated lymphomas. (**A**–**D**) Primary effusion lymphoma KSHV and EBV positive. (**A**) KSHV LANA (latency-associated nuclear antigen) staining. (**B**) Lambda light chain staining. (**C**) Kappa light staining. (**D**) EBER in situ hybridization. (**E**,**F**) Multicentric Castleman disease. (**E**) Hematoxylin-Eosin staining. (**F**) KSHV LANA staining, magnification 40×.

**Figure 6 cancers-12-02166-f006:**
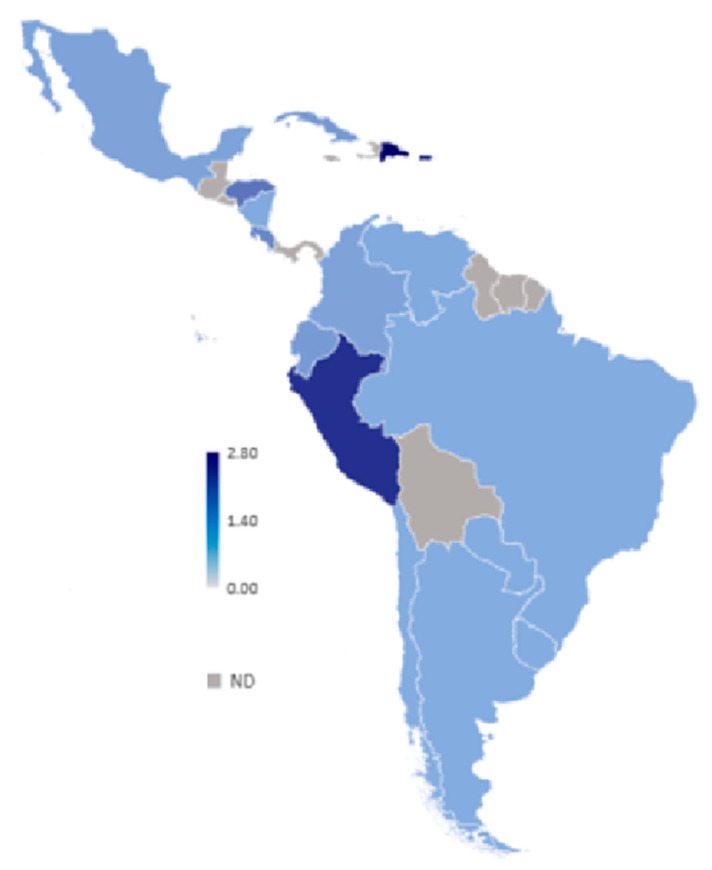
HTLV prevalence. Numeric values are in percentage. ND: not determined.

**Figure 7 cancers-12-02166-f007:**
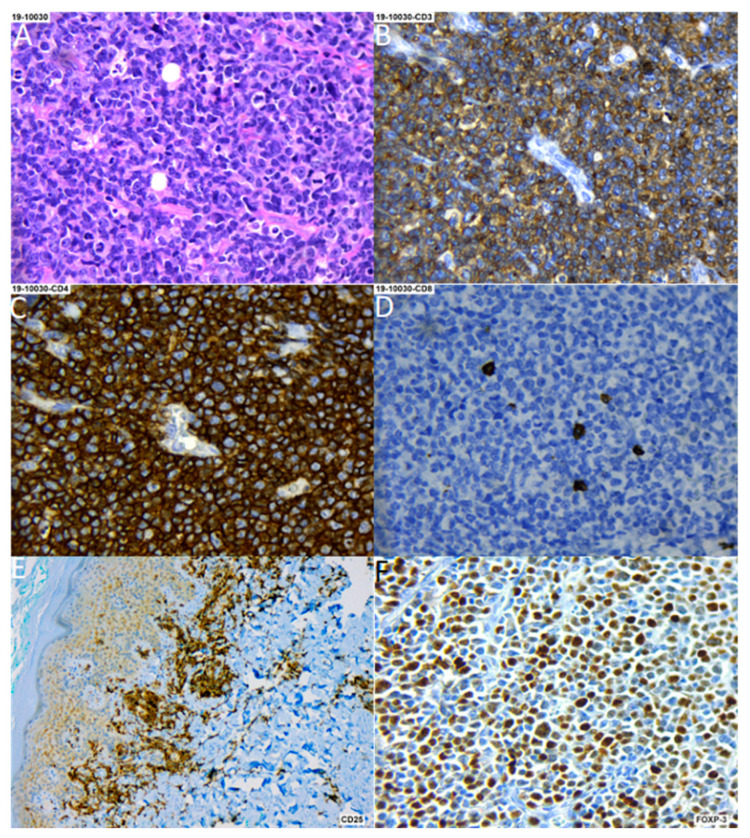
Adult T cell lymphoma (ATLL) of CD4 T cells. (**A**) Hematoxylin-Eosin staining. (**B**) CD3 staining. (**C**) CD4 staining. (**D**) CD8 staining. (**E**) CD25 staining. (**F**) FOXP3 staining. Magnification A, B, C, D and F 40×, E 10×.

**Table 1 cancers-12-02166-t001:** Summary of studies about Epstein–Barr virus (EBV) associated lymphomas.

Country (Ref)	Lymphoma Type	Type of Study	Methods	EBV Association	Description of the Study
Argentina [10,11,12]	DLBCL (ad/ped)	Cohort	ISH, IHC	12.6% (12/95)	26 DLBCL ped; 69 DLBCL ad from Buenos Aires, Argentina. EBV prevalence 72% ped.
Argentina [13]	PBL	Case report	ISH	100%	1 PBL in HIV+ patient from Buenos Aires, Argentina.
Argentina [14]	BL (ped)	Cohort	ISH, IHC	29% IHC (7/24); 100% ISH (3/3)	27 pediatric BL, 3 HIV+ from Buenos Aires, Argentina.
Argentina [15]	PBL	4 cases	ISH, IHC	100% (4/4)	4 PBL in HIV+ patients from Buenos Aires, Argentina.
Argentina [16]	B-NHL (ped)	Cohort	ISH	40% (16/40) (35% BL; 47% DLBCL)	23 BL ped, 17 DLBCL ped from Buenos Aires, Argentina.
Argentina [17]	PBL	Case report	ISH	100%	1 PBL in HIV+ from Buenos Aires, Argentina.
Argentina [18]	PBL	Case report	ISH	100%	1 PBL in HIV+ from Buenos Aires, Argentina.
Argentina [19]	T-NHL (ped)	Cohort	ISH, IHC	8% (2/25)	16 lymphoblastic, 8 anaplastic, 1 hepatoesplenic T-cell lymphoma ped from Buenos Aires, Argentina.
Argentina/Brazil [20]	HL (ped)	Cohort	ISH, IHC	54% (Arg; 60/111); 48% (Bra; 31/65)	111 HL ped from Buenos Aires, Argentina. 65 HL ped from Rio de Janeiro, Brazil.
Argentina [21]	PCNSL	7 cases	ISH, PCR	100% (6/6)	7 PCNSL cases HIV+ from Buenos Aires, Argentina.
Argentina [22]	HL (ad, ped)	Cohort	ISH	55% ped (51/92); 31% ad (25/81)	92 HL ped, 81 HL ad from Buenos Aires, Argentina.
Argentina [23]	HIV+ HL, BL, DLBCL	4 cases	ISH, IHC	100% (4/4)	1 HL, 2 BL, 1 DLBCL ped HIV+ from Buenos Aires, Argentina.
Argentina [24]	BL (ped)	Cohort	ISH, PCR	47% (8/17)	17 BL ped from Buenos Aires, Argentina.
Argentina [25,26,27]	HL (ped)	Cohort	ISH	51% (22/41)	41 HL ped from Buenos Aires, Argentina.
Argentina [28]	BL (ped)	Cohort	ISH	25% (4/16)	16 BL ped from La Plata, Argentina.
Brazil [29]	DLBCL	Cohort	ISH	30% (28/93)	93 DLBCL ad from Sao Paulo, Brazil.
Brazil [30]	NT/NKL	Case report	IHC	100%	1 NK/NKL from Rio Janeiro, Brazil.
Brazil [31]	HVL-LD	Case report	ISH	100%	1 HVL- LD from Dom Eliseu City, Pará, Brazil.
Brazil [32]	HL (ad/ped, temporal series)	Cohort	ISH	87–46% (817)	155 HL ped, 662 HL ad from Sao Paulo, Brazil.
Brazil [33]	Intermediate BL with DLBCL	Case report	ISH	100%	1 int BL DLBCL from Recife, Brazil.
Brazil [34]	DLBCL (>50 yo)	Cohort	ISH	8.45% (6/71)	71 DLBCL ad from Sao Paulo, Brazil.
Brazil [35]	T/NKL	Case report	ISH	100%	T/NKL HIV+ from Sao Paulo, Brazil.
Brazil [36]	BL (ped)	Cohort	ISH	100% (7/7)	4 BL ped, 3 BL ad from Amazonas, Brazil.
Brazil [37]	BL (ped)	Cohort	ISH	54.1% (33/61)	61 BL ped from Rio de Janeiro, Brazil.
Brazil [38]	HL (ad, ped)	Cohort	ISH	43% (56/130)	130 HL from Sao Paulo, Brazil.
Brazil [39]	HVL-LD	Case report	ISH	100%	1 HVL-LD from Manaus Amazonas, Brazil.
Brazil [40]	DLBCL, Palatine tonsil	Cohort	ISH	0% (0/26)	26 DLBCL from Bahia, Brazil.
Brazil [41,42,43]	HL (ped)	Cohort	ISH	44.8% (43/96)	96 HL ped from Rio de Janeiro, Brazil.
Brazil [44,45]	HL (ad)	Cohort	ISH	52.6% (51/97)	97 HL ad from Sao Paulo, Brazil.
Brazil [46]	ENKTCL (ad, ped)	Cohort	ISH, PCR	100% (74/74)	74 ENKTCL from Sao Paulo, Brazil.
Brazil [47]	HL (ad)	Cohort	ISH	22% (5/23)	23 HL ad from Sao Paulo, Brazil.
Brazil [48]	CNS DLBCL (ad)	Cohort	ISH	5.5% total (2/36) (40% IS)	36 CNS DLBCL from Sao Paulo, Brazil.
Brazil [49]	HL (stomach)	5 cases	ISH	80% (4/5)	5 HL from Sao Paulo, Brazil.
Brazil [50]	B-NHL (ped)	Cohort	ISH, qPCR	23% (7/30)	30 NHL ped from Rio de Janeiro and Sao Paulo, Brazil.
Brazil [51]	HL + PTL	Case report	ISH	100%	HL PTL from Sao Paulo, Brazil.
Brazil [52]	HL (ad, ped)	Cohort	ISH, IHC	50.3% (85/169)	169 HL from Sao Paulo, Brazil.
Brazil [53,54]	BL (ad, ped)	Cohort	ISH	52.6 % (123/234)	North Region(*n* = 17 cases), Central West Region(*n* = 17 cases), Northeast Region(*n* = 86 cases), Southeast Region(*n* = 72 cases), South Region(*n* = 42 cases)
Brazil [55]	BL (ped)	Cohort	ISH	66% (33/50)	143 pediatricos y 88 adultos, 3 sin edad).
Brazil [56]	BL (ped)	Cohort	ISH	61% (33/54)	San pablo
Brazil [57]	PBL	11 cases	ISH, PCR	100% (11/11)	Rio de janeiro
Brazil [58]	HL (ad, ped)	Cohort	ISH, IHC	48% (22/46)	San pablo
Brazil [59]	PCNSL	10 cases	IHC	10% (1/10)	Florianopolis, South of Brazil
Brazil [60]	HL (ad)	Cohort	IHC, PCR	37% (11/30); 43% Circulating EBV	14 patients <15 years y 32 >15 years
Brazil [61]	HL (ped)	Cohort	ISH	86.7% (78/90)	Niterói RJ
Brazil [62,63,64]	B-NHL (ped)	Cohort	ISH, PCR	72% (21/29)	29 NHL from Rio de Janeiro, Brazil.
Brazil [65]	HL (ad)	Cohort	ISH, IHC	75.6% (28/37)	37 HL ad from Ceara, Brazil.
Brazil [66]	HL (ad)	Cohort	IHC	45.8% (38/83)	83 HL ad from Rio de Janeiro, Brazil.
Brazil [67]	HL (ad, ped)	Cohort	IHC	55% (35/64)	64 HL ad from Rio de Janeiro, Brazil.
Brazil [68,69]	HL (ad)	Cohort	ISH, IHC	64.1% (50/78)	78 HL ad from Rio de Janeiro, Brazil.
Brazil [70]	HL (ad, ped)	Cohort	ISH	63.5% (61/96)	96 HL from Sao Paulo and Ceara, Brazil.
Brazil [71]	ENKTCL	Cohort	ISH	100% (16/16)	16 ENKTCL from Sao Paulo, Brazil.
Brazil [72]	BL (ped)	Cohort	ISH	73% (8/11)	11 BL ped from Recife, Brazil.
Brazil [73]	HL (ped)	Cohort	ISH	57.7% (15/26)	26 HL ped from Curitiba, Brazil.
Brazil [74]	BL (ped)	Cohort	ISH	87% (47/54)	54 BL ped from Bahia, Brazil.
Brazil [75]	BL	Cohort	ISH	71% (17/24)	24 BL from Sao Paulo, Brazil.
Brazil [76]	HL (ped)	Cohort	ISH	72% (18/25)	25 HL ped from Sao Paulo, Brazil.
Mexico [77]	HVL-LD	Cohort	ISH	100% (20/20)	20 HVL-LD ped from Mexico City, Mexico.
Mexico [78]	HL (ad, ped)	Cohort	ISH	76.1% ped (22/42); 66.6% ad (16/24)	42 HL ped, 24 HL ad from Mexico City, Mexico.
Mexico [79]	PCL	Case report	ISH	100%	1 PCL from Mexico City, Mexico.
Mexico [80]	PBL	5 cases	PCR	80% (4/5)	5 PBL ad HIV+ from Mexico City, Mexico.
Mexico [81]	DLBCL (>50 yrs)	Cohort	ISH	7% (9/136)	136 DLBCL ad from Mexico City, Mexico.
Mexico [82]	HL (mostly ad)	Cohort	ISH, IHC	61.4% (35/57)	54 HL ad, 3 HL ped from Mexico City, Mexico.
Mexico [83]	ENKTCL	Cohort	ISH	96% (22/23)	23 ENKTCL from Mexico City, Mexico.
Mexico/Bolivia [84]	HVL-LD /ACTCLC	4 cases	ISH	75% (3/4)	4 ACTCLC ped from Mexico City and La Paz
Mexico [85]	PTLD	8 cases	ISH	100% (8/8)	8 PTLD ad from Mexico City, Mexico.
Mexico [86]	NHL, Intestinal	Cohort	ISH	63% (12/19)	7 T, 6 high grade B, 6 low grade B-NHLs from Mexico City, Mexico.
Mexico [87,88]	HL (ad)	Cohort	IHC	70% (35/50)	50 HL ad from Mexico City, Mexico.
Mexico [89]	HL (ad, ped)	Cohort	ISH, IHC	67% (18/27)	3 HL ped, 24 HL ad from Mexico City, Mexico.
PR [90]	HL & NHL (ad, ped)	Cohort	IHC	50% HL (11/22); 35% NHL (22/63)	22 HL, 63 NHL from San Juan de Puerto Rico, Puerto Rico.
DR [91]	HL (ped)	Cohort	IHC	64.3% (18/28)	28 HL ped from Santiago, Dominican Republic
Cuba [92]	BL (ped)	7 cases	EBV Serology	85.7% (6/7)	EBV prevalence in 7 BL ped from La Habana, Cuba.
Colombia [93,94]	HL (ad, ped)	Cohort	ISH, IHC	67% (45/67) 60.4% ad; 84.2% ped	48 HL ad, 19 HL ped from Bogotá, Colombia
Peru [95]	DLBCL (ad)	Cohort	ISH	28% (33/117)	117 DLBCL ad from Lima and Arequipa, Peru.
Peru [96]	LBCL in cardiac myxoma (ad)	Case report	ISH	100%	LBCL ad from Lima, Peru.
Peru [97]	HVL-LD	4 cases	ISH	100% (3/3)	4 HVL-LD ped from Lima, Peru.
Peru [98]	EBV+ DLBCL GI (ad)	5 cases	ISH	100% (5/5)	5 DLBCL GI ad from Lima, Peru.
Peru [99]	Systemic T/NKL	6 cases	ISH, IHC	100% (6/6)	6 T/NKL ped from Lima, Peru.
Peru [100,101]	EBV+ DLBCL (>50 yo)	Cohort	ISH	14% (28/199)	199 DLBCL ad from Lima, Peru.
Peru [102]	DLCBL in a HTLV-1+	Case report	ISH	100%	DLBCL ad from Lima, Peru.
Peru [103]	Cutaneous T/NKL (11/15 HVL-LD)	Cohort	ISH	100% (15/15)	12 T/NKL ped, 2 T/NKL ad from Lima, Peru.
Peru [104]	ENKTCL	Cohort	PCR	99% (76/77)	77 ENKTCL from Lima, Peru.
Peru [105]	HVL-LD	6 cases	ISH	100% (6/6)	6 HVL-LD ped from Lima, Peru.
Peru [106]	ENKTCL	Cohort	ISH, IHC	96% (27/28)	28 ENKTCL ad from Lima, Peru.
Peru [107]	Nasal lymphoma	Cohort	ISH	93% (14/13; 11/11 T-cell)	13 Nasal lymphoma from Lima, Peru.
Peru [108]	HL (mostly ped)	Cohort	ISH, IHC	94% (30/32)	32 HL from Lima, Peru.
Ecuador [109]	HVL-LD	2 cases	ISH	100%	1 HVL-LD ped, 1 HVL-LD ad from Quito, Ecuador.
Ecuador [110]	HL and NHL (ad)	Cohort	ISH, qPCR	55.5% HL (5/9); 59.5% NHL (25/42)	9 HL ad, 42 NHL ad from Guayaquil, Ecuador.
Chile [111]	EBV+ DLBCL	Case report	ISH	100% (Leukocytoclastic vasculitis)	DLBCL from Santiago, Chile.
Chile [112]	Nasal lymphoma	Cohort	ISH	DLCBL 0% (0/3); T-cell 0% (0/1); NKT 100% (6/6)	3 DLBCL, 1 T-cell lymphoma, 6 NKT from Valdivia, Chile.
Chile [113]	ENKTCL	Cohort	ISH	78% (7/9)	9 ENKTCL ad from Santiago, Chile.
Chile /Argentina /Brazil [114]	BL (mostly ped)	Cohort	ISH	41% (7/17 Arg); 50% (5/10 Chile); 58% (7/12 Br)	37 BL ped, 2 BL ad from Buenos Aires, Argentina; Santiago, Chile; Campinas, Brazil.
Honduras [115]	HL (ped)	Cohort	ISH	100% (11/11)	11 HL ped from Tegucigalpa, Honduras.
CR [116]	HL (ad, ped)	Cohort	IHC	40% (16/40)	6 HL ped, 34 HL ad from San Jose, Costa Rica.
CR [117]	HL (ped)	Cohort	ISH, IHC	81% (34/42)	42 HL from San José, Costa Rica.
CR/Mexico [118]	HL (ad, >15 yo)	Cohort	ISH, SB	36% (5/14) CR; 77% (24/31) Mexico	45 HL ad from Mexico City, Mexico; San Jose, Costa Rica.
Bolivia [119]	NHL	8 cases	ISH	75% (6/8)	8 NHL from Santa Cruz, Bolivia.
Guatemala [120]	ENKTCL	Cohort	ISH	87% (73/84)	59 ENKTCL from Guatemala City, Guatemala.

ACTCLC: angiocentric cutaneous T-cell lymphomas of childhood; B-NHL: B-cell non-Hodgkin lymphoma; CNS: central nervous system; CSF: cerebrospinal fluid; DLBCL: diffuse large B cell lymphoma; GI, gastrointestinal; HL, Hodgkin lymphoma; HVL-LD: hydroa vacciniform-like lymphoproliferative disease; IC: immunocompetent; IHC: immunohistochemistry; IS: immunosuppressed; ISH: Epstein–Barr virus-encoded small RNAs (EBERs) in situ hybridization; ENKTCL: extranodal nasal-type T-cell/NK lymphoma, PBL: plasmablastic lymphoma; PCL: plasma cell myeloma; PCNSL: primary central nervous system lymphoma; PTL: peripheral T cell lymphoma; qPCR: quantitative real-time PCR; SB: Southern blot; T-NHL: T-cell Non-Hodgkin Lymphoma; Pt: patient; ad: adult; ped: pediatric. Countries PR, DR and CR: Puerto Rico, Dominican Republic and Costa Rica. virus+ refers to virus positive samples.

**Table 2 cancers-12-02166-t002:** Summary of studies about KSHV.

Country (Ref)	Type of Study	Methods	Results	Description of the Study
Brazil [148]	Prevalence	EIA, IFA	25.1% KSHV+	Screening of blood donors from Sao Paolo, Salvador and Manaos (February–December 2003). *n* = 3493
Brazil [150]	Prevalence	EIA, IFA	31.3% KSHV+	Screening of non-Amerindians blood donors from Mapuera and Manaos (Amazon). *n* = 1120
Brazil [149]	Prevalence	IFA	1.1% KSHV+ health care workers	Screening of health care workers (*n* = 757) and HIV/AIDS patients in Sao Paolo (1999–2008).
Brazil [147]	Prevalence	EIA, IFA	4.6% KSHV+	Screening of blood donors from Vitoria and Sao Paolo (January 1998–April 1999). *n* = 747
Brazil [153]	Prevalence	EIA	16.3% KSHV+	Screening of 81 low-income families from Belem Para, derived from a previous rotavirus study. *n* = 467
Brazil [145]	Prevalence	IFA	4% KSHV+	Screening of blood donors from Sao Paolo. *n* = 400
Brazil [146]	Prevalence	IFA	3.8% KSHV+	Screening of blood donors from Sao Paolo (2000–2002). *n* = 319
Brazil [152]	Prevalence	ISH	6.83% KSHV+	Screening of children (0–20 yo) from Campinas after removal of either/both tonsil or adenoids. *n* = 293
Brazil [151]	Prevalence	IFA, WB	9.35% KSHV+	Screening of children (0–12 yo) from Rio de Janeiro. *n* = 171
Brazil [154]	Case report	IFA	40 yo female with UCD hyaline variety; 62 yo with MCD mixed hyaline and plasma cell. Patients died, because of progression to NHL	Description of 3 patients with CD of which two were KSHV+
Brazil [155]	Case report	IHC, Histology	41 yo male with KSHV+ MCD co-presented with KS. 35 yo male had no KSHV test	Description of two HIV/AIDS patients with MCD plasma cell variant
Brazil [156]	Case report	IHC	32 yo male with MCD plasmablastic variant	Description of an HIV/AIDS patient with a history of KS in spite of controlled HIV load
Brazil [157]	Case report	IHC	40 yo male with MCD plasmablastic variant	Description of an HIV/AIDS patient with MCD co-existing with KS
Brazil [158]	Case report	Histology	37 yo male with MCD. No KSHV or HIV test	Description of a patient with MCD plasma cell variant
Brazil [159]	Case report	PCR	44 yo male with PEL	Description of an HIV/AIDS patient
Brazil [160]	CSS	PCR	One (2%) bone marrow sample was KSHV+	Screening of patients with multiple myeloma. *n* = 48
Arg [146]	Prevalence	IFA	3.6% KSHV+	Screening of blood donors from Buenos Aires, Bahia Blanca and Cordoba (January 2000–December 2002). *n* = 1859
Arg [161]	Prevalence	PCR	6.6% KSHV+	Screening of blood donors from Buenos Aires and San Salvador de Jujuy. *n* = 772
Arg [162]	Prevalence	EIA	7.8% KSHV+	Screening of blood donors from Buenos Aires. *n* = 577
Arg [163]	Prevalence	EIA	6.4% KSHV+	Screening of repository samples from a previous cervical cancer study from 9 countries. Concordia Argentina. *n* = 968
Arg [164]	CSS	PCR	One KSHV+	Description of DLBCLs in HIV+ patients. *n* = 11
Arg [165]	Case report	IFA, PCR	36 yo male with MCD hyaline vascular. 49 yo male MCD plasmablastic co-presenting with KS.	Report of two cases of KSHV+ MCD in HIV+ patients
Arg [13]	Case report	PCR	34 yo male with DLBCL EBV+ and KSHV+	Description of a DLBCL in an HIV/AIDS patient
Arg [166]	Case report	PCR	72 yo male HIV- with KSHV+ PEL	Description of a PEL
Arg [167]	Research study	IFA, PCR, Seq	I MCD and 1 PEL are described	KSHV genotyping
Col [163]	Prevalence	EIA	13.32% KSHV+	Screening of repository samples from a previous cervical cancer study from 9 countries. *n*= 1883 from Colombia (Bogota City)
Chile [146]	Prevalence	IFA	2.6% KSHV+	Screening of blood donors from Santiago (2000–2002). *n* = 300
Peru [168]	Prevalence	IFA	56.2% KSHV+	Screening of blood donors from one Hospital in Lima. *n* = 128
Peru [169]	CSS	IHC	Two UCD hyaline variety are reported as KSHV+. One MCD plasma cell was KSHV negative.	Description of CD patients from two hospitals in Lima. *n* = 10
Mex [170]	CSS	IHC	Only one neoplasm was a KSHV+ MCD	Description of CD patients from a Mexico City hospital (2000–2015). *n* = 39
Mex [171]	CSS	Histology	5 MCDs are found but no test for KSHV infection	Description of CD patients from a Mexico City hospital between 1996–2003. *n* = 11.
Mex [172]	CSS	Histology	1 (4.8%) PEL	Analysis of malignant neoplasms in HIV+ patients from a Mexico City hospital (January 2005–July 2008). *n* = 21
Mex [173]	Case report	IHC	73 yo HIV- female. EBV- lymphoma	Description of CD coexisting with a DLBCL.
Mex [174]	Case report	IHC	36 yo HIV+ male	Description of a KS coexisting with a plasmablastic lymphoma and previous history of CD.
PR [175]	CSS	Histology	4.8% developed neoplasia, Lymphomas are reported but are not classified further	HIV+ patients from a Bayamon Institute. *n* = 3576
Cuba [176]	Prevalence	IF, WB	1.2% KSHV+ blood donors	Screening of blood donors (*n* = 171) and risk populations.
Cuba [177]	Research Study	PCR, Seq	2 PEL patients are included. No association is found between specific KSHV strains and any disease.	Molecular classification of KSHV strains isolated from KS patients, lymph node lesions and contacts of KS patients. *n* = 90
Cuba [178]	Prevalence	IF	16.9% KSHV+	Screening of archival samples from a previous oral cancer study. *n* = 379
Cuba [179]	Case report	Histology	4 cases hyaline-vascular histological variety and 1 case to plasmacellular variety.	Description of five cases of CD
CR [163]	Prevalence	EIA	9.81% KSHV+	Screening of repository samples from a previous cervical cancer study from 9 countries. *n*= 1008 from Costa Rica (Guanacaste province)

CD: Castleman disease; CSS: cross sectional study; DLBCL: diffuse large B-cell lymphoma; EIA: enzyme linked immunoassay; HIV/AIDS: human immunodeficiency virus/acquired immunodeficiency syndrome; IF: immunofluorescence; IH: immunohistochemistry; KS: Kaposi sarcoma; MCD: multicentric Castleman disease; PA: particle agglutination; PEL: primary effusion lymphoma; RIPA: radioimmunoprecipitation; SB: Southern blot; Seq: sequencing; UCD: unicentric Castleman disease; WB: western blot, virus+ or virus- refers to virus positive or virus negative samples.

**Table 3 cancers-12-02166-t003:** KSHV in risk groups.

Country (Ref)	KSHV Prevalence Risk Groups				
	HIV+	FSW	Oral Cancer Patients	Close Contacts	MSM
Cuba [176,187]	20.8% *n* = 154		33.5% *n* = 191	46.1–72.7% X = 62.9% *n* = 35	
Brazil [149,150,188,189,190,191,192]	13.9–52.6% X = 21.8% *n* = 2358	6.7% *n* = 90			
Honduras [193]	31.8% *n* = 22	9.4% *n* = 96			
Colombia [180]	8% *n* = 25				
Peru [181]	66.5% *n* = 197				26.7% *n* = 300
Argentina [194]	17.4% *n* = 144				

Prevalence of KSHV infection in specific risk groups, percentages and the number of tested individuals (*n*) are given. When more than one study was found, the range of positives and the average (X) are also given. FSW: female sex workers, MSM: men that have sex with men.

**Table 4 cancers-12-02166-t004:** Summary of HTLV studies.

Country	Type of Study	Methods	HTLV Association	Description of Study
(Ref)			(Main Finding)	
Mexico [226]	Prevalence	PA, WB	0.39% HTLV-1/2+	Screening of blood donors from Monterrey. *n* = 1017
Mexico [227]	Prevalence	EIA	2.8% HTLV+	Screening of emergency room patients from Mexico City. *n* = 909
Mexico [228]	Prevalence	EIA, PA, WB	0.3% HTLV+ healthy women	Screening of healthy women (*n* = 662) & cervical cancer patients from Yucatan.
Panama [229]	Prevalence	EIA	5% HTLV+	Screening sera from previous studies (meningitis and enterovirus). *n* = 754
Panama [230]	Case report	EIA, WB	HTLV-1+	Description of ATLL patient
Panama [231]	CSS	EIA, IFA, WB, RIPA	7 cases HTLV+	Analysis of hematologic malignancies. *n* = 136
			3 cases confirmed ATLL	
Hond [232]	Prevalence	EIA, WB, PCR	8.1% HTLV-1+ of non- mestizos	Screening of blood donors from 15 cities of the Atlantic coast (*n* = 2651)
			0.5% HTLV-1+ of mestizos	
Hond [233]	Prevalence	EIA, WB	0.3% HTLV-1+	Screening of healthy individuals from Tegucigalpa and San Pedro Sula. *n* = 899
			1.1% HTLV-2+	
Hond [234]	Case report	EIA	HTLV+	Description of ATLL patient
Nicar [235]	Technical paper	EIA, WB, PCR	0.24% HTLV-1+	Test of a new assay in blood donors. *n* = 410
Cuba [215]	Prevalence	EIA, WB, RIPA	0.27% HTLV+	Screening of blood donors. *n* = 2579
Cuba [216]	Prevalence	EIA, WB	0.99% HTLV-1+ blood donors	Screening of blood donors (*n* = 1409) & risk groups.
Cuba [214]	Prevalence	PA, EIA, IF, WB	0.25% HTLV+ in blood donors	Screening of blood donors (*n* = 2429) & patients with hematological and non-hematological disorders from 13 provinces.
			0.72% in patients	
Cuba [178]	Prevalence	EIA	4.2% HTLV-1+	Screening of archival samples from a previous oral cancer study. *n* = 379
Cuba [220]	CSS	EIA, WB, PCR	5 ATLL reported	Hematological neoplasms in 2 sentinel centers from Havana & Matanzas, (1997–2006). *n* = 1281
DR [218]	Prevalence	EIA, WB, RIPA, IFA	2.5% HTLV+ low risk group 0% Children	Screening of low (*n* = 2552) and high risk populations.
PR [219]	Prevalence	Undisclosed	0.25% HTLV+	Database search of transfusion transmitted pathogens in blood donors. *n* = 400
PR [217]	Prevalence	EIA, WB, RIPA	O.2% HTLV-1+	Archival samples from a Dengue surveillance study & blood donors from Ponce. *n* = 1881
			1% IVDU	
Venez [236]	Prevalence	EIA, WB, PCR	0.2% HTLV+	Screening of blood donors from Caracas. *n* = 23,413
Venez [237]	Prevalence	EIA	1% HTLV+ Caracas	Screening of healthy individuals. *n* = 769
			13.7% in the Amazonas	
Venez [238]	Prevalence	EIA, WB	0.58% HTLV+	Screening of healthy individuals from an immunology clinic in Maracay City. *n* = 514
Colom [239]	Prevalence	Undisclosed	0.3 % HTLV+	Databases search of Colombian blood banks (2001–2014). *n* = 5,105,159
Colom [240]	Prevalence	EIA	0.24% HTLV+	Databases search of Cali blood banks (2008–2014). *n* = 77,117
Colom [241]	Prevalence	PA, WB	0.7% HTLV-1+ children	Screening of random subjects from Tumaco. *n* = 1077
			5.3% 20 y & older	
			7.8% IVDU	
Colom [242]	Prevalence	EIA	0.3% HTLV+	Screening of blood donors from Bogota (1999–2004). *n* = 8913
Colom [243]	Prevalence	EIA, WB	0.176% HTLV+	Databases search of Medellin blood banks (2014–2018). *n* = 52,159
			HTLV-1:HTLV-2 (3:1)	
Colom [244]	Prevalence	EIA, WB	0.06% HTLV+	Database search of one Medellin blood bank (2014–2015). *n* = 14,423
			HTLV-1:HTLV-2 (8:1)	
Colom [245]	CSS	EIA, RIA, WB	4.3% HTLV+ lowland	Comparison between Coastal and high-altitude towns. *n* = 670
			0.9% upland regions	
Colomb [246]	Case report	EIA	HTLV+ ATLL	HTLV+ ATLL co-presenting with a verrucous carcinoma
Colom [247]	Case Report	EIA, WB, PCR	Clinical presentation similar to ATLL from Japan	Description of six ATLL patients from Cali.
Colom [248]	Case Report	EIA, WB	HTLV-1+ ATLL	Description of two ATLL patients
Colom [249]	Case control	EIA, WB	100% of TSP/HAM & ATLL cases were HTLV-1+	Screening for auto-antibodies in TSP/HAM (*n* = 37) and ATLL (*n* = 10) patients.
Ecuador [250]	Prevalence	EIA, IF, WB	3.5% HTLV+	Screening of random subjects from the Esmeralda Province. *n* = 227
Ecuador [251]	Prevalence	EIA, WB	2.4% HTLV+ in Afro descendant, 0% in the Chachis	Screening of random subjects from the Esmeralda Province. *n* = 142
Ecuador [252]	Prevalence	EIA, RIPA, WB	0.42% HTLV-1+ in continental regions	Screening of healthy individuals (*n* = 744) and risk groups throughout Ecuador.
			1.57% in Galapagos Island	
Peru [253]	Prevalence	EIA, WB	3.4% HTLV+	Databases search of blood banks from Abancay (2010–2015). *n* = 2895
Peru [254]	Prevalence	EIA, WB	0.9% HTLV-1+	Databases search of blood banks from Arequipa in 2005. *n* = 2732
Peru [255]	Prevalence	EIA, WB	0.3% HTLV-1+	Screening of random subjects from 24 cities. *n* = 1535
			0% HTLV-2	
Peru [256]	Prevalence	EIA	1.7% HTLV-1+	Screening of pregnant women from Lima. *n* = 2492
Peru [257]	Case report	Undisclosed	HTLV-1+	Acute liver failure as presentation of ATLL.
Peru [258]	Case report	Undisclosed	HTLV-1+	Co-presentation of ATLL and strongyloidiasis.
Peru [259]	Case report	WB	HTLV+	Description of three ATLL patients.
Peru [260]	Case report	EIA, WB	3/7 tumors EBV+	Description of seven patients with DLBCL in HTLV-1 carriers.
			HTLV provirus in tumor was not assessed	
Peru [261]	Case report	EIA HTLV-1	HTLV-1+	Description of two cases of ATLL and concomitant S. stercoralis.
Peru [262]	Case report	EIA, WB, IH	HTLV-1+	Description of four patients with gastric ATLL
Peru [263]	Case Report	EIA, PCR	HTLV-1+	Description of one ATLL patient
Chile [264]	Prevalence	EIA, IF, PCR	0.12% HTLV+	Databases search of blood banks from 12 Chilean regions (2011–2013). *n* = 694,016
			HTLV-1: HTLV-2 (7.3:1)	
Chile [265]	Prevalence	EIA, IF, PCR	0.24% HTLV-1+	Databases search of blood banks from Valdivia (May 2009–May 2010). *n* = 6237
			Female:Male (3:1)	
Chile [266]	CSS	EIA, SB, PCR	Clinical and laboratory features of HTLV-I+ ATLL	Description of ATLL patients from the Santiago area (1989–1998). *n* = 26
Chile [267]	Case report	EIA, PA, PCR	Intestinal involvement and progression into a leukemic phase	Description of one ATLL patient
Chile [268]	Case report	EIA, SB, PCR	Clinical and laboratory features of HTLV-I+ ATLL	Description of nine ATLL patients
Chile [269]	Case report	EIA, WB	Provirus was not assessed	Description of an ATLL with Reed-Sternberg cells
Chile [270]	Case report	EIA, WB, SB	HTLV-1+ ATLL	Description of three ATLL patients co-presenting with TSP/HAM
Chile [271]	Case report	EIA, CSF, SB	HTLV-1+ ATLL	ATLL case preceded by TSP/HAM
Chile [272]	Case report	EIA, SB	HTLV-1+ ATLL	Chilean immigrant in Spain with ATLL
Chile [273]	CSS	PCR, RFLP	18.2% HTLV+ (15 HTLV-1 & 1 HTLV-2)	Analysis of hematological malignancies from Valdivia. *n* = 88
Chile [274]	CSS	Undisclosed	23 neoplasia of T cells	Analysis of chronic LPD from one hospital in Santiago (1999–2001). *n* = 132
			11 HTLV-1+ ATLL	
Chile [275]	Case report	WB	HTLV+ ATLL	ATLL case in an HIV+ patient
Parag [276]	Prevalence	PA, EIA, WB, RIPA	0% HTLV+ healthy	Screening of healthy volunteers (*n* = 338) and risk groups.
Urug [277]	Prevalence	EIA	0.13% HTLV+	Data from the National Registry Report of blood banks, 2012–2014. *n* = 297,371
Urug [278]	Prevalence	PA, IFA, WB, RIPA	0.75% HTLV-1+	Screening of blood donors (*n* = 266) and risk groups.
Urug [279]	Case report	Undisclosed	HTLV-1+	ATLL of a Peruvian immigrant co-presenting with multiple parasitosis.
Arg [280]	Prevalence	PA, EIA, WB	0.9% HTLV-1+	Databases search of blood banks from Jujuy city. *n* = 14,228
			0.04% HTLV-2+	
Arg [281]	Prevalence	EIA, PA, IFA, WB	0.05% HTLV-1+	Databases search of blood banks from Buenos Aires. *n* = 12,891
			0.03% HTLV-2+	
Arg [282]	Prevalence	EIA, PA, WB, Phylogenetic study	0.011% HTLV-1+	Databases search of blood banks from Corrientes. *n* = 9422
			0.021% HTLV-2+ 1	
			HTLV-1 Subtype A	
			HTLV-2 Subtype B	
Arg [283]	Prevalence	EIA, PA	0.11% HTLV+	Database search of a blood bank from Cordoba. *n* = 20,210
Arg [284]	CSS	EIA, PA, WB	0.028% HTLV-1+	Databases search of blood banks from 9 different provinces. *n* = 123,233
			0,025% HTLV-2+	
Arg [285]	Prevalence	PA, IF, WB, PCR	0.19% HTLV-1/2+	Databases search of blood banks from Buenos Aires. *n* = 2050
Arg [286]	Prevalence	PA, WB	0.06% HTLV-1+	Databases search of blood banks from Buenos Aires. *n* = 12,846
Arg [287]	Prevalence	PA, WB	0.036% HTLV-1+	Databases search of blood banks from Buenos Aires. *n* = 19,426
			0.01% HTLV-2+	
Arg [288]	Prevalence	PA, WB	0.035% HTLV-1	Databases search of blood banks from Buenos Aires. *n* = 28,897
			0.007% HTLV-2	
Arg [289]	Prevalence	PA, IF	0.26% HTLV-1	Databases search of blood banks from Cordoba. *n* = 5476
Arg [290]	Prevalence	PA, IF, WB	0% HTLV-1	Databases search of blood banks from Santa Fe and Santiago del Estero. *n* = 1327
Arg [291]	Prevalence	PA, EIA, WB, PCR	0.033% HTLV-1+	Databases search of blood banks from Buenos Aires. *n* = 76,246
			0.013% HTLV-2+	
Arg [292]	Prevalence	Undisclosed	1.0% HTLV+ Jujuy	Databases search of blood banks from Northern and Center regions. *n* = 130,599
			0.7% HTLV+ Salta	
			0.6% HTLV+ Formosa	
Arg [293]	Prevalence	PA, WB, PCR	0.034% HTLV-1+	Databases search of blood banks from Buenos Aires. *n* = 86,238
			0.014% HTLV-2+	
Arg [294]	Prevalence	PA, WB	0.03% HTLV-1+	Databases search of blood banks from Santa Fe. *n* = 9425
			0.05% HTLV-2+	
Arg [295]	Prevalence	EIA, PA, WB	0.00072% HTLV+	Databases search of blood banks from Misiones. *n* = 6912
Arg [296]	Prevalence	EIA	0.07% HTLV-1+	Databases search of blood banks from Buenos Aires. *n* = 28,483
			0.03% HTLV-2+	
Arg [297]	Prevalence	EIA	0.191% HTLV+	Screening of pregnant women from Córdoba. *n* = 3143
Arg [298]	Prevalence	PVT, IH, Proviral DNA integration	5 cases HTLV-1+ from native American: ATLL	34 cases of lymphoma resembling ATLL
Arg [299]	Case Report	Undisclosed	HTLV-1+ of Guinean origin	Association of B-cell lymphoma and T-cell lymphoma in HTLV-1 infection.
			NHL B and one T without phenotype of ATLL	
Arg [300]	Case report	Undisclosed	HTLV-1+	ATLL cases
	5 cases			
Arg [301]	2 cases report	Undisclosed	HTLV-1+	ATLL from 2 asymptomatic brothers
Brazil [302]	Prevalence	EIA, PCR	0.1% HTLV+	Databases search of blood banks from Riberao Preto, Sao Paulo. *n* = 301,400
			HTLV-1:2 (2:1)	
Brazil [303]	Prevalence	EIA, WB	0.04% HTLV+ in South	Databases search of blood banks from 27 urban cities. *n* = 6,218,619
			1% in Northeast	
Brazil [304]	Prevalence	EIA, WB, PCR	1.76% HTLV-1+ Subtype A	Databases search of blood banks from Salvador. *n* = 1385
Brazil [305]	Prevalence	EIA, WB	0.08%-0.2% HTLV+	Databases search of blood banks from Sao Paulo, Minas Gerais & Pernambuco. *n* = 281,760
Brazil [306]	Prevalence	EIA	0.63% HTLV+	Databases search of blood banks from Southwest Bahia. *n* = 34,400
Brazil [307]	Prevalence	EIA	0.09% HTLV+	Databases search of blood banks from Goiás. *n* = 137,209
Brazil [308]	Prevalence	EIA	0.77% HTLV-1+	Databases search of blood banks from Salvador Bahia. *n* = 233,876
			0.025% HTLV-2+	
Brazil [309]	Prevalence	PCR	1.4% HTLV-1	Databases search of blood banks from Belém, Pará. *n* = 1059
			0.5% HTLV-2	
Brazil [310]	Prevalence	EIA, PCR	0.059% HTLV-1+	Databases search of blood banks from Piauí. *n* = 37,306
			0.037% HTLV-2+	
			HTLV-1 subgroup A	
Brazil [311]	Prevalence	EIA, WB	0.13% HTLV+	Databases search of blood banks from Manaus. *n* = 87,402
Brazil [312]	Prevalence	EIA	0.58% HTLV+	Databases search of blood banks from Marajó Island. *n* = 1899
Brazil [313]	Prevalence	EIA	1.48% HTLV-1+	Databases search of blood banks from Salvador Bahía. *n* = 3451
Brazil [314]	Prevalence	EIA	0.019% HTLV-1+	Databases search of blood banks from Fortaleza. *n* = 679,610
			0.005% HTLV-2+	
Brazil [315]	Prevalence	EIA	0.1% HTLV+	Databases search of blood banks from Minas Gerais. *n* = 3,249,944
Brazil [316]	Prevalence	EIA	0.02% HTLV+	Databases search of blood banks from Uberata. *n* = 147,489
Brazil [317]	Prevalence	EIA	0.1% HTLV-1	Databases search of blood banks from Belo Horizonte. *n* = 422,600
			0.002% HTLV-2	
Brazil [318]	CSS	EIA	0.02% HTLV+	Databases search of blood banks from Riberao Preto. *n* = 377,243
Brazil [319]	Prevalence	EIA, WB	0.15% HTLV+	Databases search of blood banks from Maranhao. *n* = 365,564
Brazil [320]	Prevalence	EIA, WB	0.66% HTLV+	Databases search of blood banks from Rio Branco. *n* = 11121
Brazil [321]	Prevalence	EIA	0.3% HTLV+	Databases search of blood banks from Sao Paulo. *n* = 351,639
Brazil [322]	Prevalence	EIA, WB	0.17% HTLV+	Databases search of blood banks from Sao Paulo. *n* = 17063
Brazil [323]	CSS	Undisclosed	83 ATLL (26 smoldering, 23 chronic, 16 acute, 13 lymphoma and five primary cutaneous)	ATLL treatment and response in Salvador, Bahia
Brazil [324]	CSS	EIA, WB, PCR, IH	HTLV-1+	52 cases ATLL from Salvador, Bahía.
Brazil [325]	Case report	EIA, WB, IH	HTLV-1 in CSF + ATLL	Simultaneous development of ATLL and HAM/TSP in a pediatric patient with IDH.
Brazil [326]	CSS	EIA, WB	HTLV+ ATLL	Prognostic Factors of 72 ATLL.
Brazil [327]	Case report	EIA, WB, Nested PCR	HTLV-1+ ATLL with monoclonal integration.	ATLL case
Brazil [328]	Case report	EIA, WB, PCR	HTLV-1+, four with p16 deletion, one with p53 mutation	8 cases of pediatric ATLL patients
		FC, Cytogenetic		
Brazil [329]	CSS	EIA WB, Nested PCR	HTLV-1+ found in 48 ATLL.	188 patients with a T-cell disorder. 53 ATLL. Rio de Janeiro, Brazil
Brazil [330]	Cohort	EIA, PA, WB, Sequencing	HTLV-1+ found in 11 cases.	14 cases of ATLL, Rio de Janeiro.
Brazil [331]	Case report	EIA	HTLV-1+	Hodgkin-like ATLL from Salvador, Bahía.
Brazil [332]	Case report	EIA	HTLV-1+	ATLL from Brasilia.
Brazil [333]	Case report	EIA	HTLV-1+	ATLL with Hyalohyphomycosis
Brazil [334]	Case report	EIA	HTLV-2+	Two HTLV-2+ ATLL cases in Maranhao, Northeast Brazil.
Brazil [335]	Case report	EIA	HTLV-1+	ATLL in a patient with Strongyloides stercoralis.
Brazil [336]	Case report	EIA	HTLV-1+	IDH progressing to ATL.
Brazil [337]	Case report	EIA	HTLV-1+	Primary cutaneous type of ATLL.
Brazil [338]	Case report	EIA	HTLV-1+	2 ATLL cases from Minas Gerais.
Brazil [339]	Case report	EIA	HTLV-1+	ATLL case.
Brazil [340]	CSS	EIA, WB, PCR	HTLV-1+	195 ATLL cases from Rio de Janeiro.
			9% also with HAM/TSP	
Brazil [341]	CSS	EIA, PCR	HTLV-1+	28 ATLL from Bahía.

ATLL adult T leukemia/lymphoma; CSS: cross sectional study; DLBCL: diffuse large B-cell lymphoma; EIA: enzyme linked immunoassay; FC: flow cytometry; HIV: human immunodeficiency virus; HSCS HTLV-I/II-associated spinocerebellar syndrome; IF: immunofluorescence; IDH: infective dermatitis associated with HTLV; IDU intravenous drug users; IH: immunohistochemistry; LG: lymphomatoid granulomatosis; MF: mycosis fungoides; PA: particle agglutination; PTCL: peripheral T-cell lymphoma; RFLP restriction fragment length polymorphisms; RIPA: radioimmunoprecipitation; SB: Southern blot; SS: Strongyloides Stercoralis infection; TSP/HAM: Tropical spastic paraparesis/HTLV associated myelopathy; WB: western blot.

**Table 5 cancers-12-02166-t005:** HTLV in risk groups.

Country (Ref)	HTLV Prevalence Risk Groups							
	HIV+	STD	FSW	MSM	Cervical Cancer	IVDU	Blood Transfusion	Close Contacts
Brazil [356,365,367,368,370,371,373,376,391,392,393,394,395,396,397,398,399,400,401,402,403,404,405,406,407,408]	1.1–16.3% NT = 531/8154 (6.5%) *n* = 96 (H-1) *n* = 69 (H-2)	0.5% NT = 2/395 *n* = 2 (H-1)	1.4–3.2% NT = 31/1336 (2.3%) *n* = 30 (H-1) *n* = 1 (H-2)	0.7–4.7% NT = 38/1871 (2%) *n* = 9 (H-1)		11.7–35.2% NT = 165/854 (19.3%) *n* = 78 (H-1) *n* = 28 (H-2)	7–10.2% NT = 41/422 (9.7%) *n* = 41 (H-1)	21.2–35.1% NT = 159/517 (30.7%) *n* = 159 (H-1)
Argentina [387,409,410,411,412,413,414,415,416,417,418]	1.6–27.7% NT = 114/1158 (9.8%) *n* = 59 (H-1) *n* = 18 (H-2)	1% NT = 4/400	1.5–1.6% NT = 17/1069 (1.6%) *n* = 6 (H-1) *n* = 1 (H-2)	0.5–3% NT = 26/1216 (2.1%) *n* = 2 (H-1)		1.5–33.6% NT = 149/1128 (13.2%) *n* = 69 (H-1) *n* = 47 (H-2)		31.5% NT = 29/92 *n* = 29 (H-1)
Mexico [228,343,344,346,347,419,420,421,422]	0–12.5% NT = 31/642 (4.8%) *n* = 4 (H-1) *n* = 24 (H-2)		0–1.8% NT = 7/462 (1.5%) *n* = 0 (H-1) *n* = 5 (H-2)	0–0.9% NT = 1/239 (0.4%)	1.4–6.5% NT = 9/195 (4.6%)	20.7% NT = 106	0–0.8% NT = 3/834 (0.36%)	
DR [218,224]	5.5% NT = 6/108	2.8% NT = 14/494	4.2% NT = 13/308			2–14.3% NT = 11/120 (9.2%)		
Cuba [216,423]	0% NT = 0/268	1.7% NT = 25/1444 *n* = 25 (H-1)					10% NT = 29/921 *n* = 29 (H-1)	
Venezuela [424]			0% NT = 0/141	1% NT = 1/100 *n* = 1 (H-1)				
Ecuador [252]		0.6% NT = 1/159 *n* = 1 (H-1)	1.4% NT = 2/141 *n* = 2 (H-1)					
CR [348]					0.7% NT = 3/436 *n* = 3 (H-1)			
Colombia [241]						7.8% NT = 13/167 *n* = 13 (H-1)		
Peru [425]			9.5% NT = 184/1938 *n* = 184 (H-1)					

Prevalence of HTLV infection in specific risk groups, the range of positives and the value of the number of positives/total of tested individuals (NT) are given as percentages. Data based on one single study does not include a range. When the study addressed the HTLV type, the number of positives (*n*) found for HTLV-1 (H-1) or HTLV-2 (H-2) are also given. FSW: female sex workers, STD: sexually transmitted diseases, MSM: men that have sex with men, IVDU: intravenous drug users.

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
