# Peer review of "Lymphotropic Viruses EBV, KSHV and HTLV in Latin America: Epidemiology and Associated Malignancies. A Literature-Based Study by the RIAL-CYTED"

_cancers, 2020, doi:10.3390/cancers12082166_

Round 1
Reviewer 1 Report
The manuscript is a review by the RIAL CYTED network, the literature-based study of lymphotropic viruses, EBV, KSHV, and HTLV, epidemiology and associated malignancies in Latin America. The RIAL CYTED network aims to increase knowledge of virus-associated lymphomas, to improve the diagnosis and prognosis of these neoplasms, particularly for under-developed or developing regions throughout Latin America. In this review, the authors analyzed the literature related to the lymphotropic and lymphomagenic viruses, EBV, KSHV, and HTLV-1, in order to better understand the regional distribution of these viruses and the risk to develop lymphoma for Latin America population. The web search to access all related scientific publications in English, Portuguese, and Spanish was carried out from February 2019 to December 2019. All collected publications were classified in order to estimate these viruses association with lymphoma in specific regions of Latin America. Case reports and series of lymphomas were included and the methods of viral diagnosis were assessed.
The data in the sections, describing the viruses and the virus-associated lymphomas/diseases throughout Latin America countries and regions, are presented well, clear, and professional.
However, I have critical comments regarding to the Abstract, Introduction, Discussion, and Conclusion parts.
Since the manuscript is submitted for a scientific publication (not at all as a report for government authorities) the applied terminology must be scientifically correct.
- There are no EBV-lymphoma or KSHV-lymphoma (in Abstract: lines 38 and 41, respectively; in Table 2, in Conclusins: line 706), as well as EBV-cHL or EBV-BL (in Discussion: lines 616, 619, 626; and621, 624, 626) - see the last WHO Classification of Tumors of Hematopoietic and Lymphoid Tissues.
Suggestion: Instead should be used EBV-associated and KSHV-associated or EBV-positive and KSHV-positive. Alternatively, the authors should indicate at the first-time usage in the text like: “EBV-associated lymphoma (further, EBV-lymphoma)….etc.”
- Although the molecular and genetic studies “strongly supports involvement of EBV in the process of lymphomagenesis” of the EBV-positive classical Hodgkin lymphoma (cHL) and EBV-positive diffuse large B-cell lymphoma (DLBCL) not otherwise specified (NOS), the induction of these lymphomas by EBV is not proven but assumed (see the review by Shannon-Lowe, Rickinson, and Bell, Epstein-Barr virus-associated lymphomas., 2017 – the reference No 3 in your manuscript).
Suggestion: “EBV-induced Hodgkin lymphoma (HL) and….” (in Abstract: line 39; in Conclusions: line 707) should be replaced by the correct one (as EBV-associated, or EBV-driven, or EBV-involved, or EBV-linked, or EBV-contributed, or EBV-related).
- In the Introduction part, two references are only applied.
Suggestion (to avoid accusations in plagiarism): (1) For the information from The International Agency for Research on Cancer (in Introduction, paragraph 2), the reference to the website or publication should be given;
(2) For the information in Introduction, paragraphs 3 and 4, the references should be given (as: reviewed in A, B, and C; or: referred below). However, I am not sure that “referred below” is acceptable by the journal style.
- Typing discordance: In Introduction and Chapter 2.: “socio-economic” (lines: 92, 125). However, starting from the chapter 2.1 (lines: 177, 179, 619, 627) – there are “socioeconomic” writing.
Author Response
EBV, KSHV, and HTLV, epidemiology and associated malignancies in Latin America. The RIAL CYTED network aims to increase knowledge of virus-associated lymphomas, to improve the diagnosis and prognosis of these neoplasms, particularly for under-developed or developing regions throughout Latin America. In this review, the authors analyzed the literature related to the lymphotropic and lymphomagenic viruses, EBV, KSHV, and HTLV-1, in order to better understand the regional distribution of these viruses and the risk to develop lymphoma for Latin America population. The web search to access all related scientific publications in English, Portuguese, and Spanish was carried out from February 2019 to December 2019. All collected publications were classified in order to estimate these viruses association with lymphoma in specific regions of Latin America. Case reports and series of lymphomas were included and the methods of viral diagnosis were assessed.
The data in the sections, describing the viruses and the virus-associated lymphomas/diseases throughout Latin America countries and regions, are presented well, clear, and professional.
However, I have critical comments regarding to the Abstract, Introduction, Discussion, and Conclusion parts.
Since the manuscript is submitted for a scientific publication (not at all as a report for government authorities) the applied terminology must be scientifically correct.
- There are no EBV-lymphoma or KSHV-lymphoma (in Abstract: lines 38 and 41, respectively; in Table 2, in Conclusins: line 706), as well as EBV-cHL or EBV-BL (in Discussion: lines 616, 619, 626; and 621, 624, 626) - see the last WHO Classification of Tumors of Hematopoietic and Lymphoid Tissues.
Suggestion: Instead should be used EBV-associated and KSHV-associated or EBV-positive and KSHV-positive. Alternatively, the authors should indicate at the first-time usage in the text like: “EBV-associated lymphoma (further, EBV-lymphoma)….etc.”
Answer: We have modified the manuscript changing all mentions of EBV-lymphoma for EBV-positive lymphoma. We also changed EBV-induced lymphoma, and KSHV-lymphoma. Note that in tables EBV- lymphoma means EBV negative lymphoma. Because of space restriction in the table, we did not change it there, but note a space between – and lymphoma. We also added in the table footnote the meaning of EBV+ or EBV- as positive and negative.
- Although the molecular and genetic studies “strongly supports involvement of EBV in the process of lymphomagenesis” of the EBV-positive classical Hodgkin lymphoma (cHL) and EBV-positive diffuse large B-cell lymphoma (DLBCL) not otherwise specified (NOS), the induction of these lymphomas by EBV is not proven but assumed (see the review by Shannon-Lowe, Rickinson, and Bell, Epstein-Barr virus-associated lymphomas., 2017 – the reference No 3 in your manuscript).
Suggestion: “EBV-induced Hodgkin lymphoma (HL) and….” (in Abstract: line 39; in Conclusions: line 707) should be replaced by the correct one (as EBV-associated, or EBV-driven, or EBV-involved, or EBV-linked, or EBV-contributed, or EBV-related).
Answer: See answer to concern no. 1
- In the Introduction part, two references are only applied.
Suggestion (to avoid accusations in plagiarism): (1) For the information from The International Agency for Research on Cancer (in Introduction, paragraph 2), the reference to the website or publication should be given;
(2) For the information in Introduction, paragraphs 3 and 4, the references should be given (as: reviewed in A, B, and C; or: referred below). However, I am not sure that “referred below” is acceptable by the journal style.
Answer: We have added the IARC website. Since most of the sentences are common knowledge that often is extended later on the review, we only added one new reference and place another reference about tumor viruses throughout this introductory part. We also added (see below) when the topic will be later discussed in the manuscript.
- Typing discordance: In Introduction and Chapter 2.: “socio-economic” (lines: 92, 125). However, starting from the chapter 2.1 (lines: 177, 179, 619, 627) – there are “socioeconomic” writing.
Answer: We have homogenized the text using socio-economic.
Reviewer 2 Report
The review of Paola Chabay et al., titled “Lymphotropic viruses EBV, KSHV and HTLV in Latin America: Epidemiology and Associated malignancies. A literature-based study by the 4 RIAL-CYTED” gives an extended overview about the EBV, KSHV and HTLV associated malignancies in Latin America highlighting a high incidence of EBV-lymphomas in Mexico and Peru, of Burkitt lymphoma in North Brazil. In addition, they described an interesting high prevalence of KSHV and HTLV in Amerindian populations, which does not correlate with higher occurrence of multicentric Castleman disease, primary effusion lymphoma and adult T cell leukemia/lymphoma. On the other hands, Kaposi sarcoma has been described as the third most common neoplasm in Peru. The manuscript is well organized and it reports extensive updated literature on giving an excellent overview of the topic. Few aspects could be improved:
- Table1 on EBV associated lymphomas does not report the description of the study as it is for the table on KSHV and HTLV1, I found it very useful for the reader.
- The title of the column of the different tables are not uniform, I suggest using the same wording (e.g. Study population or description of the study? Country or Ctry?).
- I suggest describing figure 7 at the end of section 4, in the description of ATLL associated to HTLV-1 infection. Lanes 582-583 seem out of the contest described in that part of the manuscript.
- The description reported from lane 603 to 606 are relevant should be also reported in the initial description of HTLV-1 (Section 4).
- As minor comment and whether it is possible, I prefer list of abbreviations reported at the beginning of the manuscript since the reader can easily use them during the study of the manuscript.
Author Response
The review of Paola Chabay et al., titled “Lymphotropic viruses EBV, KSHV and HTLV in Latin America: Epidemiology and Associated malignancies. A literature-based study by the 4 RIAL-CYTED” gives an extended overview about the EBV, KSHV and HTLV associated malignancies in Latin America highlighting a high incidence of EBV-lymphomas in Mexico and Peru, of Burkitt lymphoma in North Brazil. In addition, they described an interesting high prevalence of KSHV and HTLV in Amerindian populations, which does not correlate with higher occurrence of multicentric Castleman disease, primary effusion lymphoma and adult T cell leukemia/lymphoma. On the other hands, Kaposi sarcoma has been described as the third most common neoplasm in Peru. The manuscript is well organized and it reports extensive updated literature on giving an excellent overview of the topic. Few aspects could be improved:
- Table1 on EBV associated lymphomas does not report the description of the study as it is for the table on KSHV and HTLV1, I found it very useful for the reader.
Answer: As the reviewer 2 correctly points out, Table 1 does not include the description of the study, as it is for KSHV and HTLV tables. This difference is based on EBV prevalence. EBV prevalence has been demonstrated worldwide, even though the age of primary infection varies in underdeveloped and developing populations compared to developed ones. However, in adults, close to 95% of people from different socioeconomic settings are seropositive to EBV. In addition, the aim of this table was to show EBV- association with different types of lymphomas in Latin America. In contrast, both KSHV and HTLV are not as prevalent, this is the reason why tables for both viruses included the columns Study Population or Description of the study, in order to indicate in detail when we referred to prevalence studies in a specific population, or to a KSHV or HTLV associated diseases. Those population prevalence studies do not apply for EBV, since EBV is widely distributed among adults, and also in children from underdeveloped and developing populations in Latin America. Under the reviewer consideration, even though our aim to describe EBV-associated diseases was different compared to KSHV and HTLV, we can include a column indicating the prevalence of EBV if indicated in the study, as well as the region where the analyses were performed.
- The title of the column of the different tables are not uniform, I suggest using the same wording (e.g. Study population or description of the study? Country or Ctry?).
Answer: We have homogenized the tables. Sometimes the name of the countries are shortened. For instance, in Table 2 and 4, but those changes were introduced by the journal editing of the manuscript
- I suggest describing figure 7 at the end of section 4, in the description of ATLL associated to HTLV-1 infection. Lanes 582-583 seem out of the contest described in that part of the manuscript.
Answer: According to Cancers Instruction for authors, all Figures, Schemes and Tables should be inserted into the main text close to their first citation and must be numbered following their number of appearance. In the paragraph above the Figure 7 we described ATLL findings, we consider that Figure 7 must remain in this place, in agreement with Cancers Instruction for authors for Figures. Concerning lines 582-583, as several publications reviewed here indicate, ATLL may have complications like cutaneous involvement, infective dermatitis (IDH), hyalohyphomycosis or Strongyloides stercoralis. We consider that it would be important to point out those findings that in our population co-present with ATLL. However, we can remove the sentence under the reviewer’s consideration.
- The description reported from lane 603 to 606 are relevant should be also reported in the initial description of HTLV-1 (Section 4).
Answer: We have moved that section to the paragraph in which we explain the HTLV-1 and HTLV-2 worldwide distribution in the HTLV introductory section.
- As minor comment and whether it is possible, I prefer list of abbreviations reported at the beginning of the manuscript since the reader can easily use them during the study of the manuscript.
Answer: We have moved the list of abbreviations after the abstracts and keywords.